# Extending Sequence Length is Not All You Need: Effective Integration of Multimodal Signals for Gene Expression Prediction

**Zhao Yang**[1,2,3,4,†*]  **Yi Duan**[1,3,4*]  **Jiwei Zhu**[1,3,4]  **Ying Ba**[1,3,4]  **Chuan Cao**[2]  **Bing Su**[1,3,4‡]

[1]Gaoling School of Artificial Intelligence, Renmin University of China, Beijing, China

[2]Zhongguancun Academy, Beijing, China

[3]Beijing Key Laboratory of Research on Large Models and Intelligent Governance

[4]Engineering Research Center of Next-Generation Intelligent Search and Recommendation, MOE

## Abstract

Gene expression prediction, which predicts mRNA expression levels from DNA sequences, presents significant challenges. Previous works often focus on extending input sequence length to locate distal enhancers, which may influence target genes from hundreds of kilobases away. Our work first reveals that for current models, long sequence modeling can decrease performance. Even carefully designed algorithms only mitigate the performance degradation caused by long sequences. Instead, we find that proximal multimodal epigenomic signals near target genes prove more essential. Hence we focus on how to better integrate these signals, which has been overlooked. We find that different signal types serve distinct biological roles, with some directly marking active regulatory elements while others reflect background chromatin patterns that may introduce confounding effects. Simple concatenation may lead models to develop spurious associations with these background patterns. To address this challenge, we propose Prism, a framework that learns multiple combinations of high-dimensional epigenomic features to represent distinct background chromatin states and uses backdoor adjustment to mitigate confounding effects. Our experimental results demonstrate that proper modeling of multimodal epigenomic signals achieves state-of-the-art performance using only short sequences for gene expression prediction.

## 1 Introduction

Understanding and predicting gene expression is fundamental to deciphering the complex regulatory mechanisms governing cellular functions (Pratapa et al., 2020). Accurate gene expression prediction enables breakthroughs across biomedicine (Mamoshina et al., 2016), from unraveling disease pathogenesis (Cookson et al., 2009; Emilsson et al., 2008) and enabling personalized therapeutic strategies (Blass & Ott, 2021), to guiding the design of specialized regulatory elements (Lal et al., 2024; Yang et al., 2025a).

However, accurately predicting gene expression presents significant challenges. First, gene expression depends on regulatory elements that can be located hundreds of thousands of base pairs (bps) away (Schoenfelder & Fraser, 2019) (Figure 1 (a)), which naturally requires models capable of processing long DNA sequences (Figure 1 (b)) (Avsec et al., 2021; Nguyen et al., 2023; Schiff et al., 2024; Su et al., 2025). Additionally, gene expression is a cell-type specific process (Shen-Orr et al., 2010) that is difficult to predict precisely using cell-shared DNA sequences alone, necessitating the integration of cell-type specific information such as histone modifications, chromatin accessibility, and other multimodal epigenomic signals (Lin et al., 2024; Su et al., 2025) (Figure 1 (c)).

---

*Equal contribution.

[†]Work was done during Zhao Yang's internship at Zhongguancun Academy.

[‡]Correspondence to: bingsu@ruc.edu.cn

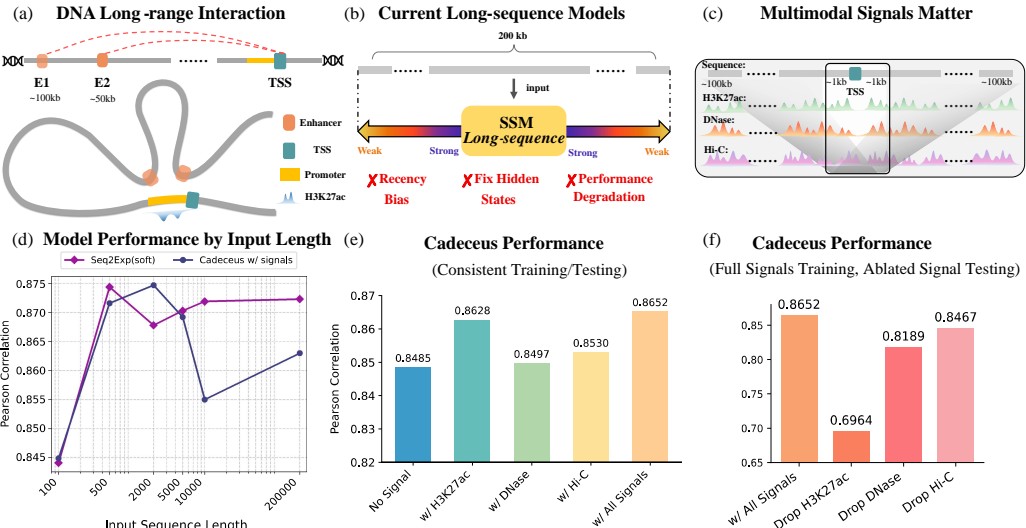

Figure 1: (a) Long-range regulatory interactions through chromatin looping. (b) Current long-sequence models suffer from technical limitations. (c) Multimodal epigenomic signals provide cell-type specific regulatory information. (d) Performance of Seq2Exp (Su et al., 2025) and Caduceus (Schiff et al., 2024) with varying input sequence lengths. (e) Different signals show varying contributions. (f) Performance degradation when specific signals are removed during testing from a model trained with all signals.

Previous works primarily focus on modeling long sequences. However, through simple but insightful experiments, we demonstrate that these methods merely mitigate the performance degradation inherent in current long-sequence modeling paradigms (Figure 1(d), details in Section 2). In contrast, using short sequences already achieves excellent results, especially when combined with multimodal epigenomic signals. We attribute the effectiveness of short sequences to the fact that proximal epigenomic signals reflect the activity of distal regulatory elements through chromatin looping and spatial interactions (Plank & Dean, 2014). As shown in Figure 1(a), although enhancers and genes are separated by large distances, some epigenomic signals near the gene can reveal the regulatory influence of these distal elements.

State-of-the-art (SOTA) methods (Lin et al., 2024; Schiff et al., 2024; Su et al., 2025) utilize epigenomic signals through simple concatenation (Figure 1(c)) without considering their distinct biological roles. We conducted a study characterizing the differential contributions of various epigenomic signals by training Caduceus (Schiff et al., 2024) with DNA sequence alone and with individual signals (H3K27ac, DNase-seq, Hi-C) or all combined. Figure 1(e) shows each signal improves performance, with H3K27ac providing the most substantial enhancement. This aligns with biological understanding: H3K27ac directly marks active regulatory elements (Creyghton et al., 2010), functioning as a *foreground* signal, while DNase-seq and Hi-C serve as *background* signals indicating chromatin accessibility (Thurman et al., 2012) and organization (Rao et al., 2014). Models trained on all signals performed comparably to H3K27ac alone, indicating background signals provide limited incremental benefit beyond foreground signals.

Figure 1(f) reveals a critical paradox: removing background signals during testing from models trained on all signals causes severe performance degradation. While background signals provide minimal standalone improvement, models develop over-dependence during training. This asymmetric behavior indicates these background patterns introduce confounding effects. The underlying mechanism stems from spurious correlations in training data, where gene expression systematically co-occurs with open chromatin patterns, causing models to learn non-causal associations between accessibility and expression levels. However, gene expression can occur independently of chromatin accessibility (Volpe et al., 2002), and our case study (Appendix D) demonstrates high expression in regions with limited accessibility, substantiating the spurious correlation hypothesis.

To address these confounding effects, we propose a simple yet effective approach, Prism (**P**roximal **r**egulatory **i**ntegration of **s**ignals for **m**RNA expression levels prediction), that learns multiple combinations of high-dimensional epigenomic features to represent distinct background chromatin states (Qiang et al., 2022). Each learned combination corresponds to a specific background state. We then apply backdoor adjustment (Pearl, 2009) to perform causal intervention across these states, thereby mitigating confounding effects and enhancing the model's predictive performance.

We summarize our contributions here:

- We challenge current approaches that use long sequence modeling for gene expression prediction, which, while biologically plausible, may not yield improvements due to limitations of present technical tools.

- We systematically analyze the differential roles of various epigenomic signals and identify that background chromatin patterns may introduce confounding effects, leading models to learn spurious associations.

- From a causal perspective, we propose Prism, an approach that learns high-dimensional feature combinations to represent background chromatin states and applies backdoor adjustment to mitigate confounding effects.

- Through extensive experimentation, we demonstrate the effectiveness of our approach, achieving state-of-the-art performance using only short sequences through a simple and effective method.

## 2 CURRENT METHODS DO NOT BENEFIT FROM LONG SEQUENCE INPUT

Mainstream deep learning methods for gene expression prediction focus on extending model input length. However, since the context length that can influence gene expression is extremely long (up to 1M bps (Avsec et al., 2025)), quadratic-complexity Transformers cannot handle such sequences. Therefore, previous works have adopted alternative approaches beyond traditional Transformers, primarily falling into two categories.

The first category comprises CNN-Transformer hybrid models, which first downsample long sequences into low-resolution bins through convolutional neural networks (CNNs), then employ Transformers to model these low-resolution bins (Avsec et al., 2021; Linder et al., 2025; Avsec et al., 2025). These works follow Enformer (Avsec et al., 2021) in performing 128-fold downsampling, resulting in the loss of single-nucleotide resolution, which is sub-optimal for DNA data where single-base variations (Avsec et al., 2025) can have profound biological impacts. Although Enformer performs well in multi-task prediction, Su et al. (2025) revealed that it underperforms compared to single-nucleotide modeling approaches like Caduceus (Schiff et al., 2024) on specialized gene expression prediction tasks. Similarly, recent work focusing on personalized gene expression prediction (Li et al., 2025) demonstrated that these approaches perform worse than Caduceus when predicting gene expression in unseen individuals. Therefore, we conclude that for gene expression prediction tasks with specialized training data, maintaining single-nucleotide resolution is crucial.

Another class leverages neural networks with linear complexity, primarily the recently popular state space models (SSMs) (Gu & Dao, 2023; Nguyen et al., 2023; Schiff et al., 2024; Nguyen et al., 2024), which directly model long sequences at single-nucleotide resolution. Recently, Seq2Exp (Su et al., 2025) achieved SOTA results in gene expression prediction by introducing learnable masks on top of Caduceus (Schiff et al., 2024), whose motivation is to learn to focus SSMs on important regulatory elements, pushing SSM-based methods to SOTA performance.

In this work, we first challenge the prevalent approach of using linear-complexity SSMs for single-nucleotide resolution long sequence modeling (Schiff et al., 2024; Su et al., 2025). These methods typically evaluate their effectiveness on long sequences only. For instance, Seq2Exp (Su et al., 2025) tested exclusively on 200K-length sequences and demonstrated superior performance over existing methods, thereby claiming enhanced long-sequence modeling capabilities. However, current SSMs merely offer computational efficiency advantages with linear complexity when processing long sequences, while their actual modeling performance remains questionable (Figure 1 (b)). Specifically, (1) SSMs have fixed-size hidden states (Gu & Dao, 2023), making it difficult to memorize all information in long sequences. (2) Wang et al. (2025) indicates that SSMs exhibit a strong recency

bias, meaning tokens in the sequence primarily interact with their nearby context. This contradicts the goal of gene expression prediction, which aims to model the relationship between target genes and distant regulatory elements.

Hence, we conducted a preliminary study to validate whether SSMs can truly handle long sequences effectively. Specifically, we trained Caduceus (Schiff et al., 2024) and Seq2Exp (Su et al., 2025) with varying input lengths centered at the transcription start site (TSS) for gene expression prediction, completely following the experimental settings of Su et al. (2025) except for sequence length. According to Figure 1 (d), we observe that Caduceus's performance consistently declines after input lengths exceed 2k. Seq2Exp, despite its carefully designed learning-to-mask mechanism for filtering unimportant regions, doesn't show a clear downward trend, but its performance with 200k input length remains essentially comparable to using just 500 bps. Figure 2 (raw data in Table 4) demonstrates that the Seq2Exp model trained on 200k sequences maintains nearly identical performance even when input sequences are shortened to 2.5k during the testing phase, suggesting that even Seq2Exp

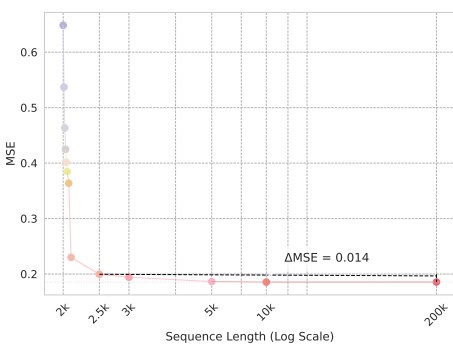

Figure 2: Shortening input length at test time.

trained on long sequences fundamentally relies only on proximal information. Therefore, rather than extending sequence length, we focus on better leveraging multimodal epigenomic signals—a longstanding overlooked direction for enhancing prediction performance.

## 3 METHOD

### 3.1 PROBLEM FORMULATION

Given a gene sequence $X = [x_1, x_2, \ldots, x_L]$, where for each $i \in \{1, 2, \ldots, L\}$, $x_i \in \mathbb{R}^4$ represents the one-hot encoding of a nucleotide base from the set $V = \{A, T, C, G\}$, and $L$ denotes the sequence length surrounding the gene's TSS (Lin et al., 2024; Su et al., 2025). For each $X$, there are associated multimodal epigenomic signals $S = [s_1, s_2, \ldots, s_L]$, where $s_i \in \mathbb{R}^d$ with $d$ representing the number of epigenomic signals. Our approach first employs a signal encoder $g_\theta : \mathbb{R}^{L \times d} \to \mathbb{R}^{L \times d'}$ with parameters $\theta$ to map the raw epigenomic signals $S$ into a higher-dimensional feature space $H = g_\theta(S)$, where $d'$ represents the dimensionality of this enriched representation following (Su et al., 2025). We then use a predictor network $h_\phi : (\mathbb{R}^{L \times 4}, \mathbb{R}^{L \times d'}) \to \mathbb{R}$ with parameters $\phi$ that integrates both sequence information $X$ and encoded epigenomic features $H$ to predict gene expression levels $Y \in \mathbb{R}$. To optimize our model parameters $\{\theta, \phi\}$, we define the following objective function:

$$\mathcal{L}_1 = \ell_{\mathrm{H}}(h_\phi(X, g_\theta(S)), Y), \tag{1}$$

where $\ell_{\mathrm{H}}$ denotes the smooth L1 loss (Huber loss) following Su et al. (2025).

### 3.2 STRUCTURAL CAUSAL MODEL

From the previous analysis, we observed that models may learn spurious associations with background epigenomic signals. To conceptualize this confounding issue, we formalize the problem using a Structural Causal Model (SCM) shown in Figure 3, where nodes represent data variables and directed edges represent hypothesized relationships. For clarity, we omit $X$ from the graph, though our model ultimately uses both $X$ and $H$.

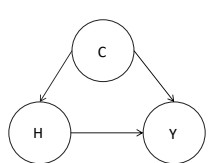

Figure 3: The SCM.

We first explain our definition of confounder $C$. In Section 1, we categorize H3K27ac as foreground signal and DNase-seq/Hi-C as background signals based on biological priors. However, this categorization is overly simplistic. H3K27ac alone cannot fully capture causal effects, as incorporating additional signals improves performance (Figure 1(e)). Similarly, background signals cannot be directly defined as confounders. Instead, we define the confounder as a more abstract concept: background chromatin

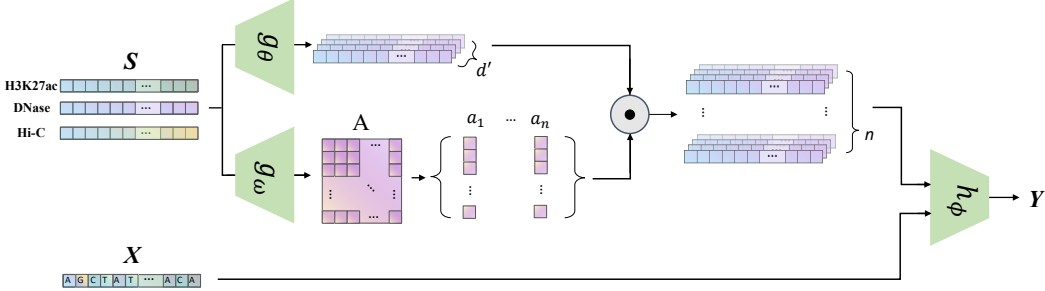

Figure 4: **Architecture of Prism.** Epigenomic signals $S$ are processed by two encoders: a signal encoder $g_\theta$ extracts high-dimension epigenomic features $H$, while a confounder encoder $g_\omega$ learns $n$ distinct weights representing the confounder $C$. A final predictor $h_\phi$ uses these weighted features along with the DNA sequence $X$ to make a prediction.

states, which represent complex combinations of multiple epigenomic signals. This aligns with approaches like ChromHMM (Ernst & Kellis, 2017), which use combinatorial patterns of epigenomic signals to define chromatin states across the genome. The specific functional implementation of $C$ is detailed in Subsection 3.4.

This definition is inspired by works in computer vision (Zhou et al., 2016; Yue et al., 2020; Qiang et al., 2022), where confounders (representing image background information) are modeled as combinations of high-dimensional semantic feature representations. Specifically, RGB images (analogous to our raw signals $S \in \mathbb{R}^{L \times d}$) are encoded into high-dimensional spaces (analogous to our $H \in \mathbb{R}^{L \times d'}$), where different linear combinations of features can represent various background contexts (Zhou et al., 2016; Qiang et al., 2022). Next, we explain the meaning of the edges in our SCM (Figure 3).

$H \to Y$. High-dimensional epigenomic features $H$ contain comprehensive regulatory information that directly influences gene expression $Y$.

$H \leftarrow C \to Y$. The confounding pathway where background chromatin state $C$ simultaneously affects both the observed epigenomic features $H$ and expression levels $Y$. For instance, globally active chromatin regions often exhibit both high accessibility signals and high expression, creating correlations that may not reflect gene-specific regulation directly.

### 3.3 CAUSAL INTERVENTION VIA BACKDOOR ADJUSTMENT

An effective prediction model should capture the direct regulatory relationship $H \to Y$ rather than spurious correlations through the confounding pathway $H \leftarrow C \to Y$. However, standard approaches optimize $P(Y|H)$, which conflates both pathways. Our goal is to estimate the interventional distribution $P(Y|do(H))$ (Pearl et al., 2016) that isolates the direct causal effect by controlling for background chromatin states $C$. The $do$ operator represents an intervention that sets $H$ while removing its dependency on confounders, enabling isolation of the direct causal effect. We stratify the confounder $C$ into $n$ distinct background chromatin states: $C = \{C_1, C_2, ..., C_n\}$, where $n$ is a hyperparameter. Using backdoor adjustment, we formulate: $P(Y|do(H)) = \sum_{i=1}^{n} P(Y|H, C = C_i)P(C = C_i)$. For computational tractability, we assume $C$ follows a uniform distribution (Qiang et al., 2022): $P(C = C_i) = \frac{1}{n}$.

### 3.4 FUNCTIONAL IMPLEMENTATION

To functionally instantiate the confounder $C$, we draw inspiration from methods in computer vision that model background context using learnable weights (Qiang et al., 2022). We introduce the confounder encoder $g_\omega : \mathbb{R}^{L \times d} \to \mathbb{R}^{n \times d'}$ with parameters $\omega$, which processes the raw epigenomic signals $S$ to generate a set of learnable weight vectors $A = [a_1, a_2, \ldots, a_n]$. Each vector $a_i \in \mathbb{R}^{d'}$ represents a distinct background chromatin state $C_i$ by applying a unique weighting scheme across the $d'$ dimensions of the encoded epigenomic features. These weights are gene-wise rather than position-wise, reflecting the assumption that background regulatory patterns are consistent across a

given gene region. For example, one weight vector might learn to emphasize chromatin accessibility signals, while another might prioritize features related to 3D chromatin organization.

This data-driven approach allows the model to capture the complex nature of background confounding effects without relying on overly simplistic biological priors. With this implementation, we can compute the interventional distribution from the backdoor adjustment formula by stratifying across these learned background states. Since we assume the DNA sequence $X$ is independent of the epigenomic features $H$ (Su et al., 2025), we include it directly in the predictor:

$$\hat{Y}_{\text{do}} = P(Y|X, do(H)) = \sum_{i=1}^{n} P(Y|X, H, C = C_i)P(C = C_i) = \frac{1}{n}\sum_{i=1}^{n} h_\phi(X, H \odot a_i), \quad (2)$$

where $\odot$ denotes element-wise multiplication. Each term $h_\phi(X, H \odot a_i)$ represents a prediction under a specific background context $C_i$. More details about the functional implementation of $h_\phi$, including the signal-sequence fusion strategy and the interventional prediction computation, are provided in Appendix F.3.

We incorporate this interventional prediction as a regularization term (Qiang et al., 2022), forming a second loss component that encourages the model to be robust to different background chromatin states:

$$\mathcal{L}_2 = \ell_{\text{H}}\left(\frac{1}{n}\sum_{i=1}^{n} h_\phi(X, H \odot a_i), Y\right). \quad (3)$$

## 3.5 TRAINING OBJECTIVE

To ensure our model learns a meaningful and diverse set of background chromatin states, we should prevent the weight vectors $\{a_i\}$ from collapsing into a single pattern. We introduce a uniform loss function (Wang & Isola, 2020) that encourages the weight vectors to be distinct from each other. This loss penalizes similarity between background representations, promoting diversity in the learned weights:

$$\mathcal{L}_3 = \log\left(\sum_{i,j} \exp\left(2t \cdot \tilde{a}_i^T \tilde{a}_j - 2t\right)\right), \quad (4)$$

where $\tilde{a}_i = a_i/\|a_i\|_2$ is the L2-normalized weight vector and $t$ is a temperature parameter that controls the sharpness of the penalty.

Our final training objective combines the standard prediction loss, the intervention-based regularization, and the uniform diversity loss:

$$\mathcal{L} = \mathcal{L}_1 + \alpha\mathcal{L}_2 + \beta\mathcal{L}_3, \quad (5)$$

where $\alpha$ and $\beta$ are hyperparameters controlling the relative importance of the intervention regularization and the uniform diversity constraint, respectively. The complete algorithm workflow for our Prism framework is provided in Appendix E.

## 4 EXPERIMENTS

### 4.1 EXPERIMENTAL SETUP

**Datasets**. To evaluate gene expression prediction, we adopt Cap Analysis of Gene Expression (CAGE) values as our prediction proxy, in line with established approaches (Avsec et al., 2021; Lin et al., 2024; Su et al., 2025). Our study focuses on two well-characterized human cell lines: K562 and GM12878. We use CAGE measurements obtained from the ENCODE (Consortium et al., 2012). Following the experimental framework established in previous studies (Lin et al., 2024; Su et al., 2025), we evaluate our model across 18,377 protein-coding genes.

For input data, we utilize both DNA sequences and epigenomic signals. The DNA sequences are derived from the human genome HG38 project, while the epigenomic signals were carefully selected (Su et al., 2025) to capture different aspects of gene regulation: **H3K27ac** marks histone acetylation at active enhancers and promoters. **DNase-seq** measures chromatin accessibility in

genomic regions, often coinciding with but not causally determining regulatory elements. **Hi-C** quantifies contact frequencies between genomic positions and the target TSS, processed using the ABC pipeline (Fulco et al., 2019). Like DNase-seq, we categorize Hi-C as a background signal representing the broader chromatin environment rather than specific regulatory elements.

Furthermore, we incorporate additional features such as mRNA half-life and promoter activity, which are taken from previous studies (Lin et al., 2024; Su et al., 2025). These features are simply concatenated to the final linear predictor and are not part of our core modeling approach for epigenomic signals.

**Baselines.** We benchmark our Prism against the following baselines: Enformer (Avsec et al., 2021), a CNN-Transformer hybrid architecture designed to predict epigenomic signals and gene expression from sequences, here used solely for CAGE prediction; HyenaDNA (Nguyen et al., 2023), Mamba (Gu & Dao, 2023), and Caduceus (Schiff et al., 2024), three recently developed DNA foundation models leveraging efficient long-sequence modeling capabilities through SSMs as prediction backbones; EPInformer (Lin et al., 2024), which extends the Activity-By-Contact (ABC) model (Fulco et al., 2019) by utilizing DNase-seq peaks to define potential regulatory regions and applying attention mechanisms to aggregate enhancer signals; and Seq2Exp (Su et al., 2025), a recent SOTA method that applies information bottleneck principles to learn regulatory element masks, available in hard (binary) and soft (continuous) variants. We also include Caduceus w/signal, which incorporates epigenomic signals directly into Caduceus's encoder, and MACS3 (Zhang et al., 2008), which differs from Seq2Exp by using MACS3-identified regulatory elements instead of learned masks. Most baseline models process raw DNA sequences from the input region, while EPInformer operates on potential enhancer candidates extracted based on DNase-seq measurements following the ABC model (Fulco et al., 2019).

**Evaluation Metrics.** We assess model performance using three metrics following Su et al. (2025): Mean Squared Error (MSE) for measuring prediction variance with emphasis on larger errors; Mean Absolute Error (MAE) for quantifying average prediction deviation in expression units; and Pearson Correlation for evaluating how well models capture expression patterns and gene rankings regardless of absolute scale. These metrics together provide a balanced assessment of both prediction accuracy and pattern preservation capabilities.

**Implementation Details.** We partition datasets by chromosome for training, validation, and testing, following Su et al. (2025). Specifically, chromosomes 3 and 21 serve as the validation set, while chromosomes 22 and X are reserved for testing. The inclusion of chromosome X provides a more stringent evaluation of model robustness due to its distinct biological characteristics compared to autosomes.

Our signal encoder $g_\theta$ is implemented as a simple linear layer (Su et al., 2025), while the confounder encoder $g_\omega$ utilizes a lightweight 1D-CNN, with details in Appendix F. For the predictor $h_\phi$, we adopt Caduceus (Schiff et al., 2024) as our backbone model, following Seq2Exp (Su et al., 2025). Notably, we maintain the same training hyperparameters as in Seq2Exp (Su et al., 2025). Further performance gains could likely be achieved through hyperparameter fine-tuning specific to our approach. We use the smooth L1 loss (Huber loss) as our prediction loss function, while the best model is selected based on the MSE metric on the validation set following Su et al. (2025). All experiments were conducted on NVIDIA A40 and A100 GPUs. While most baseline models process inputs of length 200k, our Prism implementation operates on sequences of 2k bps. Additional implementation details can be found in Appendix F.

## 4.2 RESULTS OF GENE EXPRESSION PREDICTION

Table 1 presents performance results across all methods for the K562 and GM12878 cell types, respectively. All baseline results are directly cited from Seq2Exp (Su et al., 2025) to ensure fair comparison. Additionally, all results reported include the mean and standard deviation from five runs using different random seeds: {2, 22, 222, 2222, 22222} following Su et al. (2025). The best-performing method for each metric is highlighted in bold, with the second-best underlined. Notably, our Prism consistently outperforms the previous SOTA Seq2Exp-soft across all datasets and metrics. Among the six total metrics, only K562's MAE and Pearson correlation show improvements less than one standard deviation, while all other metrics demonstrate robust improvements exceeding

Table 1: Performance on Gene Expression CAGE Prediction with Standard Deviation for Both Cell Types.

| | K562 | | | GM12878 | | |
|---|---|---|---|---|---|---|
| | MSE ↓ | MAE ↓ | Pearson ↑ | MSE ↓ | MAE ↓ | Pearson ↑ |
| Enformer | $0.2920 \pm 0.0050$ | $0.4056 \pm 0.0040$ | $0.7961 \pm 0.0019$ | $0.2889 \pm 0.0009$ | $0.4185 \pm 0.0013$ | $0.8327 \pm 0.0025$ |
| HyenaDNA | $0.2265 \pm 0.0013$ | $0.3497 \pm 0.0012$ | $0.8425 \pm 0.0008$ | $0.2217 \pm 0.0018$ | $0.3562 \pm 0.0012$ | $0.8729 \pm 0.0010$ |
| Mamba | $0.2241 \pm 0.0027$ | $0.3416 \pm 0.0026$ | $0.8412 \pm 0.0021$ | $0.2145 \pm 0.0021$ | $0.3446 \pm 0.0022$ | $0.8788 \pm 0.0011$ |
| Caduceus | $0.2197 \pm 0.0038$ | $0.3327 \pm 0.0070$ | $0.8475 \pm 0.0014$ | $0.2124 \pm 0.0037$ | $0.3436 \pm 0.0031$ | $0.8819 \pm 0.0009$ |
| EPInformer | $0.2140 \pm 0.0042$ | $0.3291 \pm 0.0031$ | $0.8473 \pm 0.0017$ | $0.1975 \pm 0.0031$ | $0.3246 \pm 0.0025$ | $0.8907 \pm 0.0011$ |
| MACS3 | $0.2195 \pm 0.0023$ | $0.3455 \pm 0.0018$ | $0.8435 \pm 0.0013$ | $0.2340 \pm 0.0028$ | $0.3654 \pm 0.0017$ | $0.8634 \pm 0.0020$ |
| Caduceus w/ signals | $0.1959 \pm 0.0036$ | $0.3187 \pm 0.0036$ | $0.8630 \pm 0.0008$ | $0.1942 \pm 0.0058$ | $0.3269 \pm 0.0048$ | $0.8928 \pm 0.0017$ |
| Seq2Exp-hard | $0.1863 \pm 0.0051$ | $0.3074 \pm 0.0036$ | $0.8682 \pm 0.0045$ | $0.1890 \pm 0.0045$ | $0.3199 \pm 0.0040$ | $0.8916 \pm 0.0027$ |
| Seq2Exp-soft | $0.1856 \pm 0.0032$ | $0.3054 \pm 0.0024$ | $0.8723 \pm 0.0012$ | $0.1873 \pm 0.0044$ | $0.3137 \pm 0.0028$ | $0.8951 \pm 0.0038$ |
| Prism | $\mathbf{0.1789} \pm 0.0041$ | $\mathbf{0.3000} \pm 0.0058$ | $\mathbf{0.8751} \pm 0.0036$ | $\mathbf{0.1759} \pm 0.0054$ | $\mathbf{0.3054} \pm 0.0048$ | $\mathbf{0.9016} \pm 0.0024$ |

Table 2: Hyperparameter sensitivity analysis for Prism on the K562 cell line. We evaluate the model's performance while varying (a) the number of background states $n$, (b) the intervention loss weight $\alpha$, and (c) the diversity loss weight $\beta$.

(a) Sensitivity on $n$

| $n$ | MSE ↓ | MAE ↓ | Pearson ↑ |
|---|---|---|---|
| 0 | $0.1863 \pm 0.0035$ | $0.3092 \pm 0.0050$ | $0.8713 \pm 0.0023$ |
| 1 | $0.1891 \pm 0.0047$ | $0.3084 \pm 0.0039$ | $0.8676 \pm 0.0032$ |
| 2 | $0.1789 \pm 0.0041$ | $0.3000 \pm 0.0058$ | $0.8751 \pm 0.0036$ |
| 3 | $0.1818 \pm 0.0091$ | $0.3018 \pm 0.0090$ | $0.8739 \pm 0.0031$ |
| 4 | $\mathbf{0.1762} \pm 0.0071$ | $\mathbf{0.2961} \pm 0.0070$ | $\mathbf{0.8780} \pm 0.0028$ |
| 5 | $0.1788 \pm 0.0062$ | $0.2996 \pm 0.0071$ | $0.8752 \pm 0.0030$ |
| 6 | $0.1857 \pm 0.0078$ | $0.3057 \pm 0.0047$ | $0.8737 \pm 0.0022$ |

(b) Sensitivity on $\alpha$

| $\alpha$ | MSE ↓ | MAE ↓ | Pearson ↑ |
|---|---|---|---|
| 0.1 | $0.1829 \pm 0.0065$ | $0.3037 \pm 0.0078$ | $0.8725 \pm 0.0030$ |
| 1.0 | $\mathbf{0.1789} \pm 0.0041$ | $\mathbf{0.3000} \pm 0.0058$ | $\mathbf{0.8751} \pm 0.0036$ |
| 10.0 | $0.1916 \pm 0.0055$ | $0.3119 \pm 0.0071$ | $0.8709 \pm 0.0029$ |

(c) Sensitivity on $\beta$

| $\beta$ | MSE ↓ | MAE ↓ | Pearson ↑ |
|---|---|---|---|
| 0.1 | $0.1789 \pm 0.0056$ | $0.2993 \pm 0.0037$ | $0.8757 \pm 0.0038$ |
| 1.0 | $\mathbf{0.1789} \pm 0.0041$ | $\mathbf{0.3000} \pm 0.0058$ | $\mathbf{0.8751} \pm 0.0036$ |
| 10.0 | $0.1836 \pm 0.0120$ | $0.3027 \pm 0.0123$ | $0.8748 \pm 0.0036$ |

one standard deviation. These results provide strong evidence that our approach achieves new SOTA performance in gene expression prediction.

### 4.3 HYPERPARAMETER SENSITIVITY ANALYSIS

Our method introduces several hyperparameters: the number of background chromatin states $n$, and coefficients $\alpha$ and $\beta$ that balance the loss components in our training objective (Equation 5). We conducted a sensitivity analysis on the K562 cell line, with results (Also averaged results from five runs, here only the mean values are shown) presented in Table 2. Our analysis of $n$ shows that while performance peaks at $n = 4$, configurations with $n \geq 2$ substantially outperform the $n = 0$ baseline, validating our intervention; we select $n = 2$ to balance performance and efficiency. For the intervention weight $\alpha$, we found performance is optimal at 1.0 and degrades when either disabled ($\alpha = 0$) or set too high ($\alpha = 10.0$). This confirms its role as an auxiliary regularizer, consistent with prior work (Qiang et al., 2022). Finally, the diversity constraint proves to be robust. The model's performance is nearly identical for $\beta = 0.1$ and $\beta = 1.0$, and shows only a slight degradation even with a large weight of $\beta = 10.0$.

### 4.4 ANALYSIS OF LEARNED WEIGHTS

To understand how our model represents the confounder $C$, we visualize the weights learned by the confounder encoder $g_\omega$ (Figure 5). The analysis reveals two key properties. First, we observe strong intra-gene diversity: for a given gene, the two learned weight vectors ($a_1$ and $a_2$) are distinct and often complementary, confirming that our model learns non-redundant representations for each confounder stratum. Second, we find evidence of inter-gene structural similarity. The overall intensity of the learned weights is clearly gene-specific, reflecting each gene's unique local epigenomic context. Despite this variation in magnitude, the relative pattern between the two states is remarkably consistent across different genes, suggesting the model learns a generalizable strategy—such as an

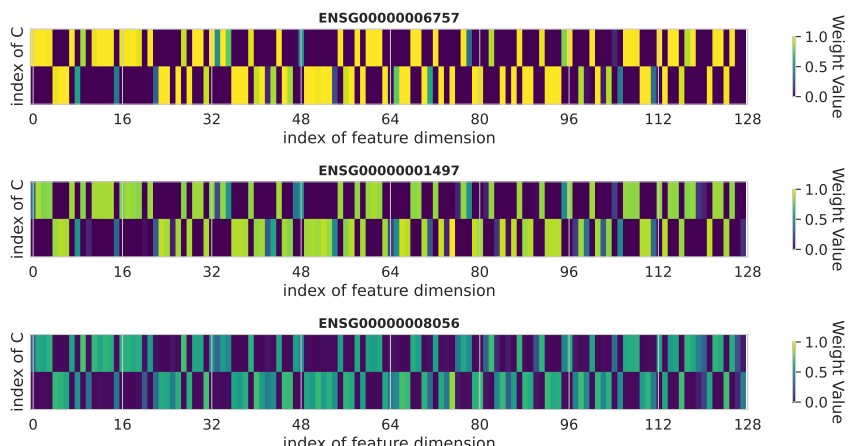

Figure 5: Visualization of learned confounder weights $(a_1, a_2)$ for three sampled genes.

"activating" versus a "suppressive" state—which it then adapts to each gene's local context. These structured representations support the validity of our causal framework.

## 4.5 PARAMETER OVERHEAD

Our confounder encoder is designed to be lightweight while delivering substantial performance improvements. We compare the additional parameters introduced by Prism and Seq2Exp (Su et al., 2025) relative to the base Caduceus (Schiff et al., 2024). As shown in Table 3, Prism adds only 11K trainable parameters to the base model. Our lightweight confounder encoder $g_\omega$ introduces minimal parameter overhead, whereas Seq2Exp's mask generator causes

Table 3: Parameter comparison between models.

| Model | Trainable Parameters |
|---------|----------------------|
| Caduceus | 574K |
| Seq2Exp | 1.1M |
| Prism | 585K |

its parameter count to double compared to Caduceus. Notably, our approach outperforms Seq2Exp across all metrics while maintaining an almost unchanged parameter count compared to Caduceus.

## 5 RELATED WORK

**Sequence-to-function models** are designed to predict functional genomic signals directly from DNA sequences. DeepSEA (Zhou & Troyanskaya, 2015) established this approach by utilizing convolutional neural networks (CNNs) to extract sequence features for multi-task prediction. The field has evolved through architectural innovations and expanded training datasets (Kelley et al., 2018; Zhou et al., 2018; Chen et al., 2022). Currently, Enformer (Avsec et al., 2021) represents the leading method, achieving exceptional performance through its hybrid Transformer-CNN architecture. While these models simultaneously predict various outputs, including epigenomic signals and gene expression levels, they typically lack specialized mechanisms for leveraging epigenomic data to enhance expression prediction, specifically, treating all prediction targets as parallel outputs rather than considering their biological interdependencies.

**Unsupervised DNA foundation models** leverage the successful paradigm of unsupervised pretraining established in natural language processing. DNABERT (Ji et al., 2021) was the first to adapt this approach to genomics, applying BERT-like Devlin et al. (2019) techniques to learn transferable DNA representations. Subsequent models have expanded upon this foundation (Zhou et al., 2024; Dalla-Torre et al., 2024; Li et al., 2024; Sanabria et al., 2024). In parallel, generative frameworks like Evo (Nguyen et al., 2024) have emerged (Nguyen et al., 2023; Brixi et al., 2025), enabling functional element design applications (Linder et al., 2025; Yang et al., 2025a). Despite these advances, such

models' effectiveness for gene expression prediction remains limited due to their exclusive reliance on DNA sequence information, without incorporating the critical epigenomic context that modulates gene activity.

**Gene expression prediction** represents a fundamental challenge in bioinformatics (Segal et al., 2002). Early approaches like Enformer (Avsec et al., 2021), Borzoi (Linder et al., 2025) and SPACE (Yang et al., 2025b) attempted to predict gene expression directly from DNA sequences, facing inherent limitations, while GraphReg (Karbalayghareh et al., 2022) enhanced performance by incorporating epigenomic information through graph attention networks to model physical interactions between genomic regions. More recent methods have progressed toward integrating both sequence and epigenomic information, with Creator (Li et al., 2023) and EPInformer (Lin et al., 2024) demonstrating improved performance through this combined approach. However, these models typically rely on pre-identified regulatory elements, overlooking potential contributions from unannotated regions. Seq2Exp (Su et al., 2025) addressed this limitation through an end-to-end, data-driven methodology that simultaneously learns to identify relevant regulatory elements and predict expression with epigenomic guidance. Despite these advances, current research tends to focus predominantly on modeling distal regulatory elements through long sequence architectures, rather than optimizing the utilization of biologically interrelated epigenomic signals that directly influence gene regulation.

## 6 CONCLUSION

This work reveals a critical challenge in gene expression prediction: while previous methods focus on modeling longer sequences, current technical paradigms suffer from inherent performance degradation with extended sequence length. Instead, we discovered that proximal epigenomic signals are crucial, but complex background chromatin states may introduce confounding effects, creating spurious correlations in models. Building on these insights, we propose Prism, a lightweight framework that achieves state-of-the-art gene expression prediction performance through effective integration of multimodal epigenomic signals using only short sequences while adding minimal computational overhead.

## ETHICS STATEMENT

We acknowledge that we have read and adhered to the ICLR Code of Ethics.

If the reviewers or the community raise any ethical concern about our work, we are ready to address them transparently and responsibly.

## REPRODUCIBILITY STATEMENT

To facilitate reproducibility, we detail the complete pipeline of our Prism framework using pseudocode in the appendix. We also provide an exhaustive documentation of the experimental setups, covering computing infrastructure, random seeds, and optimization details such as learning rate schedules and batch sizes. Furthermore, the robustness of our results is thoroughly validated through systematic evaluations, detailed ablation studies, and hyperparameter sensitivity analyses presented throughout the paper. Code is available at https://github.com/yangzhao1230/Prism.

## ACKNOWLEDGEMENTS

This work was supported in part by the National Natural Science Foundation of China No. 62376277, Public Computing Cloud, Renmin University of China, and fund for building world-class universities (disciplines) of Renmin University of China.

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

## A   THE USE OF LARGE LANGUAGE MODELS (LLMS)

We used large language models (LLMs) only to aid or polish the writing of this manuscript. They were not involved in idea generation, methodological design, experiments, or analysis. All scientific content was created and verified by the authors, who take full responsibility for the final text.

## B  Shortening Input Sequence Length at Test Time

In Figure 1 (d) of Section 1, we have confirmed that training with longer sequences from scratch does not provide additional benefits. Further, we aim to investigate whether shortening the input length at test time would decrease the performance of a model trained on longer sequences. Specifically, we tested the Seq2Exp-soft model (Su et al., 2025)[1] trained on 200k sequences to evaluate if reducing context during inference affects performance. As shown in Table 4, we found that Seq2Exp, despite being trained on 200k inputs, shows minimal performance degradation when the input length is reduced from 200K to 2.5K during testing. This suggests that Seq2Exp fails to effectively utilize long-context information even during training, indicating that the model does not genuinely leverage the extended sequence information it was provided.

Interestingly, however, there is a significant performance drop when inputs are shortened to 2,500 tokens, with a particularly sharp decline observed below 2,100 tokens. We attribute this behavior to an implementation detail in Seq2Exp: the model forcibly prevents the central 2,000-bp region from undergoing masking (this constraint was not mentioned in the Seq2Exp paper but can be found in their GitHub repository), effectively forcing the model to focus predominantly on the central 2,000 bp and proximal regulatory information.

Based on these observations and comparing with Figure 1, we can conclude that input context length has a much smaller impact on model performance than epigenomic signals. Removing epigenomic signals during testing substantially hurts performance, while shortening sequence length has minimal effect. This finding motivates our focus on modeling epigenomic information effectively.

Table 4: Performance of Seq2Exp (Su et al., 2025) when testing with shortened input sequences on the K562 cell line.

| Input Length | MSE ↓ | MAE ↓ | Pearson ↑ |
|---|---|---|---|
| 200000 | 0.1856 | 0.3054 | 0.8723 |
| 10000 | 0.1855 | 0.3074 | 0.8751 |
| 8000 | 0.1864 | 0.3082 | 0.8747 |
| 3000 | 0.1943 | 0.3134 | 0.8698 |
| 2500 | 0.1996 | 0.3174 | 0.8674 |
| 2100 | 0.2301 | 0.3471 | 0.8603 |
| 2070 | 0.3639 | 0.2464 | 0.8576 |
| 2050 | 0.3848 | 0.2686 | 0.8540 |
| 2040 | 0.4017 | 0.2855 | 0.8521 |
| 2030 | 0.4248 | 0.3093 | 0.8496 |
| 2020 | 0.4634 | 0.3543 | 0.8429 |
| 2010 | 0.5371 | 0.4506 | 0.8291 |
| 2000 | 0.6485 | 0.6183 | 0.8084 |

## C  Experimental Data of Table 1

We provide comprehensive numerical results corresponding to Figure 1 in the main text, including complete performance metrics and ablation studies.

### C.1  Sequence Length Sensitivity

Table 5 compares the performance stability of Seq2Exp and Caduceus across different input lengths.

### C.2  Epigenomic Signal Contributions

Table 6 demonstrates that combining all epigenomic signals yields optimal performance, with H3K27ac showing the strongest individual impact.

---

[1] Model available at: https://huggingface.co/xingyusu/GeneExp_Seq2Exp/tree/main

Table 5: Performance comparison with varying input lengths (left: Seq2Exp (Su et al., 2025), right: Caduceus (Schiff et al., 2024))

| Length | MAE | MSE | Pearson | | Length | MAE | MSE | Pearson |
|--------|-----|-----|---------|---|--------|-----|-----|---------|
| 100 | 0.3394 | 0.2233 | 0.8441 | | 100 | 0.3385 | 0.2200 | 0.8449 |
| 500 | 0.3096 | 0.1879 | 0.8744 | | 500 | 0.3096 | 0.1889 | 0.8716 |
| 2000 | 0.3150 | 0.1971 | 0.8678 | | 2000 | 0.3036 | 0.1831 | 0.8747 |
| 5000 | 0.3098 | 0.1949 | 0.8703 | | 5000 | 0.3170 | 0.1941 | 0.8692 |
| 10000 | 0.3088 | 0.1897 | 0.8719 | | 10000 | 0.3235 | 0.2029 | 0.8550 |

Table 6: Caduceus performance with different epigenomic signal configurations

| Configuration | MSE | MAE | Pearson r |
|---------------|-----|-----|-----------|
| No signals | 0.2163 | 0.3325 | 0.8485 |
| +H3K27ac | 0.1873 | 0.3080 | 0.8628 |
| +DNase | 0.2089 | 0.3227 | 0.8497 |
| +Hi-C | 0.2135 | 0.3264 | 0.8530 |
| All signals | 0.1886 | 0.3079 | 0.8652 |

## C.3 REMOVE SIGNALS AT TEST TIME

Table 7 reveals critical signal dependencies. Removing H3K27ac during testing from a model trained on all signals degrades performance most severely (22.3% MAE increase), while Hi-C removal has minimal effect (4.7% MAE increase).

Table 7: Performance degradation from signal removal (trained with all signals)

| Condition | MAE | MSE | Pearson |
|-----------|-----|-----|---------|
| Drop H3K27ac | 0.5653 | 0.6115 | 0.6964 |
| Drop DNase | 0.3890 | 0.2962 | 0.8189 |
| Drop Hi-C | 0.3548 | 0.2280 | 0.8467 |
| Baseline (all signals) | 0.3078 | 0.1886 | 0.8652 |

# D CASE STUDY AND QUANTITATIVE EVIDENCE OF WIDESPREAD BACKGROUND CONFOUNDERS

## D.1 QUANTITATIVE PREVALENCE OF LONG-RANGE INTERACTIONS

To statistically validate the ubiquity of long-distance chromatin interactions in the K562 and GM12878 cell lines – a core premise of our work – we conducted a two-fold analysis. This quantitative evidence establishes that the case study in Section D.2 is representative of a genome-wide signal-to-noise challenge.

First, we analyzed promoter-centric Hi-C contact data. For each gene's TSS, we examined its vector of Hi-C contact frequencies, defining a significant interaction as any contact with a signal strength greater than 0.01. We classified an interaction as "long-range" if the genomic distance from the TSS exceeded 50kb. Second, to specifically quantify connections to putative regulatory elements, we analyzed pre-computed Promoter-Enhancer (P-E) linkages from the Activity-by-Contact (ABC) model, identifying all genes connected to at least one distal enhancer (>50kb away).

The results, presented in Table 8, demonstrate that long-distance interactions are a ubiquitous feature of both genomes. Nearly all genes (∼99%) exhibit numerous long-range contacts, with a median of nearly 200,000 potential interaction partners per gene. The ABC model data further confirms that virtually all genes are linked to at least one distal enhancer. This creates a significant signal-

to-noise problem, as the vast number of interactions indicated by Hi-C data cannot all be causally determinative of gene expression, thus acting as background confounders.

Table 8: Statistical Summary of Hi-C Long-Range Interactions in K562 and GM12878 Cell Lines

| Statistic | K562 | GM12878 |
|---|---|---|
| % of genes with promoter-interacting[1] | 98.9% | 99.3% |
| % of genes with long-range (>50kb) promoter[1] | 98.9% | 99.3% |
| % of genes linked to a distal enhancer via ABC model[2] | 100.0% | 100.0% |
| Median number of long-range partners per gene[1] | 199,899 | 199,899 |
| Median distance of long-range interactions (kb)[1] | 49,707.0 | 49,939.0 |

[1]Statistics derived from promoter-centric Hi-C contact vectors.
[2]Statistics derived from pre-computed ABC model P-E links.

## D.2 QUALITATIVE CASE STUDY

The statistical prevalence of interactions motivates our hypothesis that many of these signals act as confounders rather than direct regulators. To qualitatively support this, we present a representative case (Figure 6) at a genomic locus (Entrez ID: ENSG00000080561).

In this region, both DNase (chromatin accessibility) and Hi-C (3D spatial proximity) signals exhibit broad, high activation. Despite this permissive chromatin environment, the marker of active regulatory elements, H3K27ac, shows no enrichment. Consequently, the gene expression level remains low (0.6021).

This case demonstrates that high background signal activity alone is insufficient to drive gene expression. The absence of H3K27ac indicates that key regulatory elements are inactive, resulting in minimal transcriptional output despite strong accessibility and spatial contacts.

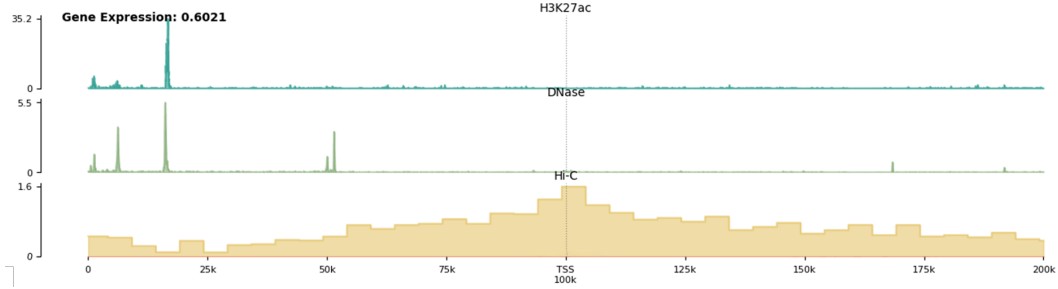

Figure 6: A representative genomic locus (Entrez ID: ENSG00000080561) where DNase and Hi-C signals are broadly active, but H3K27ac shows no enrichment. Despite strong chromatin accessibility and spatial contacts, gene expression remains low (0.6021). This supports the hypothesis that such pervasive background signals (quantified in Table 8) act as confounders rather than causal regulators.

## D.3 CONCLUSION AND MOTIVATION FOR OUR METHOD

Together, the quantitative data and the qualitative case reinforce the necessity of disambiguating causal foreground signals (like H3K27ac) from pervasive background confounders (like broad DNase and Hi-C signals) when modeling gene expression. This pervasive signal-to-noise problem directly motivates our approach of explicitly modeling background signals through a Structural Causal Model and applying backdoor adjustment to correct for their confounding effects, thereby improving both interpretability and prediction accuracy.

# E  ALGORITHM WORKFLOW

Here we provide the complete algorithm workflow for our Prism framework in Algorithm 1. The algorithm initializes three neural networks: the signal encoder $g_\theta$, the predictor network $h_\phi$, and the confounder encoder $g_\omega$. During training, we compute both standard and interventional predictions, then optimize the model using three objectives: prediction loss $\mathcal{L}_1$, intervention loss $\mathcal{L}_2$, and uniform diversity loss $\mathcal{L}_3$.

---

**Algorithm 1** Interventional Framework for Gene Expression Prediction (Prism)

---

**Require:** Gene sequence $X$, epigenomic signals $S$, gene expression $Y$, hyperparameters $\alpha, \beta, t, n$.
**Ensure:** Trained model parameters $\theta, \phi, \omega$.
  1: **Initialize** parameters $\theta, \phi, \omega$ randomly.
  2: **while** not converged **do**
  3:    *//— Forward Pass —*
  4:    $H \leftarrow g_\theta(S)$ {Encode epigenomic signals into features}
  5:    $\hat{Y} \leftarrow h_\phi(X, H)$ {Make standard prediction}
  6:    $\{a_1, \ldots, a_n\} \leftarrow g_\omega(S)$ {Learn confounder weights representing $C$}
  7:    *//— Interventional Prediction via Backdoor Adjustment —*
  8:    $\hat{Y}_{\mathrm{do}} \leftarrow \frac{1}{n} \sum_{i=1}^{n} h_\phi(X, H \odot a_i)$ {Apply backdoor adjustment}
  9:    *//— Loss Computation —*
 10:    $\mathcal{L}_1 \leftarrow \ell_{\mathrm{H}}(\hat{Y}, Y)$ {Standard prediction loss (Huber loss)}
 11:    $\mathcal{L}_2 \leftarrow \ell_{\mathrm{H}}(\hat{Y}_{\mathrm{do}}, Y)$ {Intervention loss (Huber loss)}
 12:    $\tilde{a}_i \leftarrow a_i / \|a_i\|_2$ for all $i$ {L2-normalize weight vectors}
 13:    $\mathcal{L}_3 \leftarrow \log \left( \sum_{i,j} \exp(2t \cdot \tilde{a}_i^T \tilde{a}_j - 2t) \right)$ {Uniform diversity loss}
 14:    $\mathcal{L} \leftarrow \mathcal{L}_1 + \alpha \mathcal{L}_2 + \beta \mathcal{L}_3$ {Total objective}
 15:    *//— Backward Pass —*
 16:    Update $\theta, \phi, \omega$ using gradient descent on $\mathcal{L}$.
 17: **end while**
 18: **return** $\theta, \phi, \omega$.

---

# F  MORE IMPLEMENTATION DETAILS

## F.1  TRAINING SETTINGS

Our training framework is implemented using PyTorch Lightning. All training-related hyperparameters were adopted directly from Seq2Exp (Su et al., 2025), which means we did not perform extensive parameter tuning for our specific approach. Consequently, there is potential for further performance improvements through careful hyperparameter optimization. The complete set of hyperparameters used in our experiments is presented in Table 9.

Table 9: Hyperparameter values following Seq2Exp (Su et al., 2025).

| Hyperparameters | Values |
|---|---|
| Layers of backbone | 4 |
| Hidden dimensions | 128 |
| Max training steps | 50000 |
| Batch size | 8 |
| Learning rate | 5e-4 |
| Scheduler strategy | CosineLR with Linear Warmup |
| Initial warmup learning rate | 1e-5 |
| Min learning rate | 1e-4 |
| Warmup steps | 5,000 |
| Validation model selection criterion | validation MSE |

### F.2 IMPLEMENTATION DETAILS OF CONFOUNDER ENCODER

Our confounder encoder $g_\omega$ is implemented as a lightweight 1D-CNN that maps raw epigenomic signals $S \in \mathbb{R}^{L \times d}$ to weight vectors $A \in \mathbb{R}^{n \times d'}$. The architecture consists of a three-layer CNN followed by a projection layer:

- **Layer 1:** Conv1D (in_channels=$d$, out_channels=8, kernel_size=7) followed by BatchNorm, ReLU, and MaxPool (kernel_size=4)
- **Layer 2:** Conv1D (in_channels=8, out_channels=16, kernel_size=5) followed by BatchNorm, ReLU, and MaxPool (kernel_size=4)
- **Layer 3:** Conv1D (in_channels=16, out_channels=32, kernel_size=3) followed by Batch-Norm, ReLU, and MaxPool (kernel_size=4)
- **Global Pooling:** AdaptiveAvgPool1D(1) followed by Flatten
- **Projection:** Linear layer mapping the flattened features (32 dimensions) to $n \times d'$ dimensions

The progressive reduction in sequence length through max pooling operations (by a factor of 64 in total) efficiently captures patterns at different genomic scales while significantly reducing the computational overhead. After obtaining the raw weights, we apply a sigmoid activation function to constrain the values between 0 and 1, making them suitable for weighting the epigenomic signals via the Hadamard product operation. This lightweight design adds minimal parameters to the overall model while effectively modeling the background epigenomic regulatory patterns. The entire encoder requires only 11K parameters, which is negligible compared to the backbone model's parameter count.

### F.3 IMPLEMENTATION DETAILS OF THE PREDICTOR

In practice, both the DNA sequence $X$ and the weighted epigenomic signal $H \odot a_i$ are first independently projected to the model's hidden dimension $d'$ via separate linear layers (the signal encoder $g_\theta$ and the sequence input projection, respectively), and then combined via element-wise addition before being fed into the Caduceus backbone.

Furthermore, when computing the interventional prediction $\hat{Y}_{\text{do}}$ (Equation 3), rather than averaging $n$ separate end-to-end predictions, we average the intermediate backbone representations across the $n$ contexts before applying the shared prediction head. Formally, $\hat{Y}_{\text{do}} = h_\phi^{\text{head}}\left(\frac{1}{n}\sum_{i=1}^{n} h_\phi^{\text{body}}(X, H \odot a_i)\right)$, where $h_\phi^{\text{body}}$ denotes the Caduceus backbone and $h_\phi^{\text{head}}$ denotes the prediction head. This representation averaging is a standard and computationally efficient approximation, and closely approximates the formulation in the main text, with the two being exactly equivalent when the prediction head is linear.

## G THE EFFECT OF PRE-TRAINING

Our main experiments follow Seq2Exp Su et al. (2025), where all models are trained from scratch without pre-training. To further investigate whether DNA model pre-training benefits gene expression prediction, we conducted experiments using pre-trained Enformer (pre-trained on 200k sequences) and Caduceus (pre-trained on 131k sequences) with signals, as shown in Table 10, to examine the effectiveness of long-context pre-training.

Overall, we find that pre-training provides consistent improvements. For Enformer, the improvement is marginal. For Caduceus, which was pre-trained on 131k sequences, loading pre-trained weights before fine-tuning on 200k gene expression prediction tasks yields substantial improvements, with Pearson correlation notably improving by 0.0113. However, the effect of pre-training mirrors that of Seq2Exp—it can only mitigate the performance degradation caused by extended sequence length, rather than making long-context models superior to short-context ones.

When we use only 2k input length, pre-training also provides some improvement, but this improvement is relatively modest. The absolute MSE improvement is particularly negligible. We also experimented with loading pre-trained Caduceus weights for Prism training and found that

Table 10: The effect of pre-training

| Model | Metric | From Scratch | Pre-trained |
|-------|--------|--------------|-------------|
| Enformer | MSE ↓ | $0.2920 \pm 0.0050$ | $0.2913$ (↓ $0.0007$) $\pm 0.0209$ |
| | Pearson ↑ | $0.7961 \pm 0.0019$ | $0.7983$ (↑ $0.0022$) $\pm 0.0229$ |
| Caduceus w/signals (2k input) | MSE ↓ | $0.1863 \pm 0.0035$ | $0.1858$ (↓ $0.0005$) $\pm 0.0082$ |
| | Pearson ↑ | $0.8713 \pm 0.0023$ | $0.8759$ (↑ $0.0046$) $\pm 0.0042$ |
| Caduceus w/signals (200k input) | MSE ↓ | $0.1959 \pm 0.0036$ | $0.1897$ (↓ $0.0062$) $\pm 0.0026$ |
| | Pearson ↑ | $0.8630 \pm 0.0008$ | $0.8743$ (↑ $0.0113$) $\pm 0.0030$ |
| Prism | MSE ↓ | $0.1789 \pm 0.0041$ | $0.1795$ (↑ $0.0006$) $\pm 0.0061$ |
| | Pearson ↑ | $0.8751 \pm 0.0036$ | **0.8774** (↑ $0.0023$) $\pm 0.0018$ |

it achieves stable improvements while Prism continues to maintain state-of-the-art performance. Notably, pre-training significantly improves Pearson correlation, while MSE shows minimal change.

Therefore, our conclusion is that long-context pre-training can substantially improve long-context capabilities, but this improvement only mitigates the performance degradation inherent to long-context models, while providing only marginal improvements for short-context models.

## H   EXTENDED ANALYSIS WITH ADDITIONAL EPIGENOMIC SIGNALS

To comprehensively evaluate the effectiveness of different epigenomic signals in gene expression prediction, we conducted additional experiments on the K562 cell line using signals beyond the three primary ones (H3K27ac, DNase-seq, and Hi-C) employed in our main analysis following Lin et al. (2024); Su et al. (2025).

### H.1   ADDITIONAL SIGNAL DESCRIPTIONS

We incorporated three additional epigenomic signals with distinct biological functions:

**H3K4me3** (ENCODE ID: ENCFF405ZDL): A histone modification signal that specifically marks active promoter regions with high precision, complementing H3K27ac which marks both promoters and enhancers with broader coverage.

**DNase footprint** (ENCODE ID: ENCSR000EOT): High-resolution protein-DNA binding footprints derived from DNase-seq data, identifying exact transcription factor binding sites within accessible chromatin regions through computational algorithms.

**ChIA-PET** (ENCODE ID: ENCFF278RFG): Protein-mediated chromatin interaction data that captures functionally relevant long-range contacts, providing more targeted information compared to genome-wide Hi-C interactions.

Signal processing followed standard protocols: H3K4me3 and ChIA-PET used direct bigwig signal values, while DNase footprint regions from bigbed annotations were encoded as binary signals (1 for annotated regions, 0 elsewhere).

### H.2   INDIVIDUAL SIGNAL ANALYSIS

Table 11 presents the performance of Caduceus with individual signals. Most signals demonstrate improvements over the no-signal baseline, with H3K4me3 showing the most substantial enhancement (MSE: 0.1801, Pearson: 0.8781). ChIA-PET showed degraded performance, likely due to high noise levels in the raw data. DNase footprint performed comparably to DNase-seq, suggesting limited additional information content despite higher theoretical resolution. The superior performance of H3K4me3 and H3K27ac aligns with their roles as direct indicators of active regulatory elements, supporting our categorization as foreground signals with stronger causal relationships to gene expression.

## H.3 Compositional Signal Effects

Table 12 examines the effects of combining multiple signals. Adding DNase footprint to the initial three signals provides minimal improvement, consistent with its derivation from DNase-seq data. However, incorporating H3K4me3 yields substantial performance gains across all metrics.

Most notably, Prism with H3K4me3 integration achieves the best performance (MSE: 0.1719, representing a 0.0137 improvement over Seq2Exp baseline). This demonstrates that Prism's causal intervention framework maintains robust improvements even when strong individual signals like H3K4me3 are present, suggesting that the method effectively disentangles genuine regulatory signals from confounding background effects.

## H.4 Key Findings

Our extended analysis reveals several important insights: First, signals with direct regulatory roles (H3K4me3, H3K27ac) provide greater predictive value than background accessibility signals. Second, computationally derived signals like DNase footprint offer limited additional information beyond their source data. Third, Prism consistently outperforms baseline approaches across different signal combinations, validating the robustness of our causal intervention framework. These findings support the importance of careful signal selection and highlight the potential for further improvements through strategic integration of complementary epigenomic data types.

Table 11: Single Signal Input (Caduceus w/signals (2k input))

| Signal | MSE ↓ | MAE ↓ | Pearson ↑ |
|---|---|---|---|
| No Signal | $0.2215 \pm 0.0086$ | $0.3342 \pm 0.0081$ | $0.8502 \pm 0.0026$ |
| H3K27ac | $0.1986 \pm 0.0059$ | $0.3179 \pm 0.0054$ | $0.8645 \pm 0.0037$ |
| DNase-seq | $0.2207 \pm 0.0060$ | $0.3342 \pm 0.0085$ | $0.8530 \pm 0.0037$ |
| Hi-C | $0.2202 \pm 0.0045$ | $0.3330 \pm 0.0064$ | $0.8489 \pm 0.0039$ |
| H3K4me3 | $0.1801 \pm 0.0079$ | $0.3084 \pm 0.0099$ | $0.8781 \pm 0.0018$ |
| ChIA-PET | $0.2262 \pm 0.0062$ | $0.3387 \pm 0.0060$ | $0.8422 \pm 0.0059$ |
| DNase footprint | $0.2186 \pm 0.0073$ | $0.3300 \pm 0.0045$ | $0.8523 \pm 0.0044$ |

Table 12: Effect of Compositional Signal Input (Models with 2k input)

| Model Configuration | MSE ↓ | MAE ↓ | Pearson ↑ |
|---|---|---|---|
| Caduceus w/signals (initial 3 signals) | $0.1863 \pm 0.0035$ | $0.3092 \pm 0.0050$ | $0.8713 \pm 0.0023$ |
| Caduceus w/signals (initial 3 signals + DNase footprint) | $0.1870 \pm 0.0059$ | $0.3092 \pm 0.0058$ | $0.8703 \pm 0.0026$ |
| Caduceus w/signals (initial 3 signals + H3K4me3) | $0.1789 \pm 0.0122$ | $0.3067 \pm 0.0117$ | $0.8804 \pm 0.0101$ |
| Caduceus w/signals (initial 3 signals + DNase footprint + H3K4me3) | $0.1762 \pm 0.0054$ | $0.3072 \pm 0.0031$ | $0.8837 \pm 0.0024$ |
| Prism (initial 3 signals) | $0.1789 \pm 0.0041$ | $0.3000 \pm 0.0058$ | $0.8751 \pm 0.0036$ |
| Prism (initial 3 signals + DNase footprint) | $0.1794 \pm 0.0064$ | $0.2996 \pm 0.0055$ | $0.8752 \pm 0.0042$ |
| Prism (initial 3 signals + H3K4me3) | $\mathbf{0.1719} \pm 0.0070$ | $\mathbf{0.2969} \pm 0.0049$ | $0.8839 \pm 0.0035$ |
| Prism (initial 3 signals + DNase footprint + H3K4me3) | $0.1730 \pm 0.0055$ | $0.3005 \pm 0.0051$ | $\mathbf{0.8850} \pm 0.0020$ |

## I    CROSS-CELL GENERALIZATION

Our main experiments follow EPInformer (Lin et al., 2024) and Seq2Exp (Su et al., 2025) in training separate models for each cell type. To evaluate whether a single model can generalize across cell types, we conducted a mixed-training experiment combining K562 and GM12878. Specifically, during training, for each gene we randomly sample either its K562 or GM12878 epigenomic signals and corresponding expression value. The Table 13 below shows that this mixed model achieves comparable performance to cell-type-specific models.

Table 13: Performance of mixed-training

| Dataset | Model | MSE ↓ | MAE ↓ | Pearson ↑ |
|---------|-------|-------|-------|-----------|
| K562 | Seq2Exp (cell-specific) | 0.1856 ± 0.0032 | 0.3054 ± 0.0024 | 0.8723 ± 0.0012 |
| K562 | Prism (cell-specific) | 0.1789 ± 0.0041 | 0.3000 ± 0.0058 | 0.8751 ± 0.0036 |
| K562 | Prism (mixed-training) | 0.1875 ± 0.0085 | 0.3084 ± 0.0077 | 0.8662 ± 0.0049 |
| GM12878 | Seq2Exp (cell-specific) | 0.1873 ± 0.0044 | 0.3137 ± 0.0028 | 0.8951 ± 0.0038 |
| GM12878 | Prism (cell-specific) | 0.1759 ± 0.0054 | 0.3054 ± 0.0048 | 0.9016 ± 0.0024 |
| GM12878 | Prism (mixed-training) | 0.1759 ± 0.0038 | 0.3027 ± 0.0041 | 0.9012 ± 0.0032 |

## J    PERFORMANCE ACROSS DIFFERENT INPUT LENGTHS

We have explored how performance changes with varying sequence lengths. In Table 14 below, we present the Pearson correlation results for Caduceus w/signal (Schiff et al., 2024), Seq2Exp (Su et al., 2025), and Prism on K562 across different input lengths ranging from 100 to 10,000 bp.

Table 14: Pearson across different input lengths on K562

| Input Length | Caduceus w/signal | Seq2Exp-soft | Prism ($\alpha = 1$) | Prism ($\alpha = 2$) |
|--------------|-------------------|--------------|----------------------|----------------------|
| 100 | 0.8488 ± 0.0042 | 0.8492 ± 0.0064 | 0.8493 ± 0.0056 | - |
| 500 | 0.8719 ± 0.0043 | 0.8694 ± 0.0051 | 0.8726 ± 0.0045 | - |
| 2000 | 0.8713 ± 0.0023 | 0.8643 ± 0.0088 | 0.8751 ± 0.0036 | - |
| 5000 | 0.8662 ± 0.0035 | 0.8675 ± 0.0035 | 0.8690 ± 0.0037 | - |
| 10000 | 0.8614 ± 0.0059 | 0.8699 ± 0.0032 | 0.8661 ± 0.0027 | 0.8699 ± 0.0023 |

As shown in Table 14, Prism exhibits a similar trend to Caduceus, with performance beginning to decline when input length increases beyond 2000-5000 bp. While Seq2Exp maintains relatively robust performance across lengths, their reported results at 200k only match the performance of Caduceus at 500-2000 bp, validating our claim that Seq2Exp merely mitigates the performance degradation caused by extending sequence length in Caduceus. Prism achieves the best performance among all three models across lengths from 100 to 5000 bp. We hypothesize that as input length increases, the confounding effects of background signals become stronger. To test this, we experimented with increasing the intervention loss weight to $\alpha = 2$ when the input length reaches 10,000 bp, and observed performance improvement.

## K    PERFORMANCE ON H1 CELL LINE

We conducted additional experiments on the H1 cell line (which only appeared in Seq2Exp's rebuttal stage but was not included in their final camera-ready version). The results are shown in Table 15 below.

Table 15: Performance on H1 cell line

| Method | MSE ↓ | MAE ↓ | Pearson ↑ |
|---|---|---|---|
| Caduceus w/signal (2k input) | 0.2751 ± 0.0104 | 0.3929 ± 0.0103 | 0.6681 ± 0.0137 |
| Seq2Exp-soft (our reproduction) | 0.2784 ± 0.0064 | 0.3957 ± 0.0045 | 0.6595 ± 0.0089 |
| Prism | 0.2642 ± 0.0060 | 0.3817 ± 0.0044 | 0.6844 ± 0.0078 |

On H1, Seq2Exp performs worse than the Caduceus baseline, while Prism consistently achieves SOTA results across all metrics. These results on a third cell line further demonstrate the consistent improvements of our approach.

## L  LEARNING CONFOUNDER WEIGHTS WITHOUT SUPERVISION

Unsupervised learning of chromatin states is a well-established approach in genomics, such as ChromHMM (Ernst & Kellis, 2017). ChromHMM defines states based on combinatorial patterns of epigenomic marks without explicit labels, which are subsequently mapped to biological chromatin states through expert annotation. Similarly, although our confounder weights are learned without supervision, Figure 5 in our manuscript shows that the model captures structured, gene-specific patterns rather than random noise. This suggests the model may learn meaningful latent states driven by the prediction task, which could potentially be validated with the assistance of biological experts like ChromHMM (Ernst & Kellis, 2017).

We distinguish our approach from naive data augmentation. The fundamental difference is that our weights are learnable and gene-dependent, whereas naive augmentation relies on random perturbations. While random augmentation is effective in domains like computer vision, applying it to biological signals carries the risk of destroying critical information.

To validate this distinction, we first conducted a dropout experiment on the Caduceus baseline, where input signals are randomly discarded during training. Here, we define the retention rate as the proportion of signals preserved (i.e., 1 - dropout probability). We observed that performance degrades significantly as the retention rate decreases. As shown in Table 16 below, keeping only 70% of signals results in a notable performance drop, and keeping 50% degrades the MSE further to 0.2248. In contrast, we further evaluated our approach by using the learned weights to element-wise multiply the raw high-dimensional signals for prediction. We found that the average weight values on the K562 test set is approximately 0.35, which corresponds to an average retention rate of approximately 35%. However, unlike the random dropout baseline, our model maintains robust performance despite this high sparsity. This demonstrates that the learnable weights are meaningful, selectively preserving essential biological information.

Table 16: Comparison of dropout and learned weight methods

| Model | MSE ↓ | MAE ↓ | Pearson ↑ |
|---|---|---|---|
| Caduceus w/ Dropout (rate = 0.9) | 0.1874 ± 0.0074 | 0.3062 ± 0.0064 | 0.8702 ± 0.0026 |
| Caduceus w/ Dropout (rate = 0.7) | 0.2059 ± 0.0075 | 0.3199 ± 0.0010 | 0.8625 ± 0.0041 |
| Caduceus w/ Dropout (rate = 0.5) | 0.2248 ± 0.0115 | 0.3428 ± 0.0119 | 0.8446 ± 0.0109 |
| Prism w/ Learned Weights (rate ≈ 0.35) | 0.1834 ± 0.0092 | 0.3032 ± 0.0083 | 0.8745 ± 0.0061 |

