# OpenReview forum: "Extending Sequence Length is Not All You Need: Effective Integration of Multimodal Signals for Gene Expression Prediction"
_ICLR.cc/2026/Conference — ICLR 2026 Oral_

### Official Review · Reviewer_uN9e · 2025-10-29

**Soundness:** 4
**Presentation:** 4
**Contribution:** 3
**Rating:** 6
**Confidence:** 5

**Summary:**

This paper suggests that in the problem of gene expression prediction, sequence length may not be the key factor for prediction performances. Instead, leveraging epigenomic signals can better predict the gene expression.

**Strengths:**

- The idea itself is very interesting. While many works model DNA sequences to show off their long-sequence ability, modeling long sequences in DNA itself may not be helpful to many genomics related works.
- The experiments are comprehensive to support the paper claims.

**Weaknesses:**

While the proposed $L_2$ objective is conceptually appealing, it is not fully clear how the model can learn meaningful confounder weights $a_i$ without any external supervision, but only based on epigenomic signals $S$. I can understand the performance gain from data augmentation perspective (e.g., to avoid overfit), but it’s not fully clear to me how the model can get the real confounder. Could the author elaborate it more clearly?

**Questions:**

The following points are intended as open discussions
- Following the weakness part, how do the authors think that adding additional external information could help the model learn more interpretable confounder representations $a_i$?
- The current analysis only focuses on one task: gene expression prediction. I wonder whether the authors think the same causal regularization principles could generalize to broader DNA signal modeling tasks?. Would the authors expect similar trends to hold in those settings?

---

> ### Author Response · Authors · 2025-11-21
> **Response to Reviewer uN9e （Part 1/2)**
>
> Thank you for your comments and helpful suggestions. We address your concerns in detail below.
>
> >W1: It is not fully clear how the model can learn meaningful confounder weights without any external supervision. I can understand the performance gain from data augmentation perspective (e.g., to avoid overfit), but it’s not fully clear to me how the model can get the real confounder.
>
> Unsupervised learning of chromatin states is a well-established approach in genomics, such as ChromHMM [1]. ChromHMM defines states based on combinatorial patterns of epigenomic marks without explicit labels, which are subsequently mapped to biological chromatin states through expert annotation. Similarly, although our confounder weights are learned without supervision, Figure 5 in our manuscript shows that the model captures structured, gene-specific patterns rather than random noise. This suggests the model may learn meaningful latent states driven by the prediction task, which could potentially be validated with the assistance of biological experts like ChromHMM [1].
>
> We distinguish our approach from naive data augmentation. The fundamental difference is that our weights are learnable and gene-dependent, whereas naive augmentation relies on random perturbations. While random augmentation is effective in domains like computer vision, applying it to biological signals carries the risk of destroying critical information.
>
> To validate this distinction, we first conducted a dropout experiment on the Caduceus baseline, where input signals are randomly discarded during training. Here, we define the retention rate as the proportion of signals preserved (i.e., 1 - dropout probability). We observed that performance degrades significantly as the retention rate decreases. As shown in the table below, keeping only 70% of signals results in a notable performance drop, and keeping 50% degrades the MSE further to 0.2248. In contrast, we further evaluated our approach by using the learned weights to element-wise multiply the raw high-dimensional signals for prediction. We found that the average weight values on the K562 test set is approximately 0.35, which corresponds to an average retention rate of approximately 35%. However, unlike the random dropout baseline, our model maintains robust performance despite this high sparsity. This demonstrates that the learnable weights are meaningful, selectively preserving essential biological information.
>
> | Model | MSE ↓ | MAE ↓ | Pearson ↑ |
> | --- | --- | --- | --- |
> | Caduceus w/ Dropout (retention rate = 0.9) | 0.1874 ± 0.0074 | 0.3062 ± 0.0064 | 0.8702 ± 0.0026 |
> | Caduceus w/ Dropout (retention rate = 0.7) | 0.2059 ± 0.0075 | 0.3199 ± 0.0010 | 0.8625 ± 0.0041 |
> | Caduceus w/ Dropout (retention rate = 0.5) | 0.2248 ± 0.0115 | 0.3428 ± 0.0119 | 0.8446 ± 0.0109 |
> | Prism w/ Learned Weights (retention rate ≈ 0.35) | **0.1834 ± 0.0092** | **0.3032 ± 0.0083** | **0.8745 ± 0.0061** |

---

> ### Author Response · Authors · 2025-11-21
> **Response to Reviewer uN9e （Part 2/2)**
>
> > Q1: Following the weakness part, how do the authors think that adding additional external information could help the model learn more interpretable confounder representations $a_i$?
>
> One promising direction would be to leverage single-cell epigenomic data to identify representative chromatin state prototypes. Bulk data averages signals across heterogeneous cell populations, limiting our ability to observe diverse chromatin states. Single-cell data can reveal richer chromatin variation by capturing cell-to-cell heterogeneity. We could identify genes with similar expression patterns across single cells, extract their chromatin states, and cluster these chromatin profiles to define biologically meaningful prototypes. These prototypes could serve as weak supervision through an auxiliary loss that aligns our learned confounder weights with observed chromatin patterns.
>
> > Q2: I wonder whether the authors think the same causal regularization principles could generalize to broader DNA signal modeling tasks? Would the authors expect similar trends to hold in those settings?
>
> Thank you for this valuable question. We believe causal-inspired methods hold substantial promise for genomic deep learning. Currently, most genomic deep learning approaches directly adapt techniques from natural language processing (NLP). However, a critical difference between DNA and natural language is that DNA has much lower information density, making correlation-based deep learning particularly prone to learning spurious correlation.
>
> We expect that similar causal principles could be valuable for variant effect prediction (VEP), particularly for non-coding variants, which represents a key goal of genomic modeling. The relevance comes from a well-known challenge in genetics: linkage disequilibrium (LD), where multiple genetic variants tend to co-occur, yet only a subset is truly causal for the observed effects.
>
> To our knowledge, no existing work has successfully applied end-to-end modern deep learning to identify causal non-coding variant effects. Notable efforts such as Borzoi [2] and AlphaGenome [3] rely heavily on strong biological priors (e.g., predefined exon annotations and manually designed scoring schemes) rather than learning causal relationships in an end-to-end manner from genomic data. We believe causal-inspired frameworks could help address this challenge.
>
> **References**:
>
> [1] Ernst et al., ChromHMM: automating chromatin-state discovery and characterization, Nature Methods, 2012
>
> [2] Linder et al., Predicting RNA-seq coverage from DNA sequence as a unifying model of gene regulation, Nature Genetics, 2025
>
> [3] Avsec et al., AlphaGenome: Advancing regulatory variant effect prediction with a unified DNA sequence model, bioRxiv, 2025

---

### Official Review · Reviewer_oNqu · 2025-10-30

**Soundness:** 3
**Presentation:** 3
**Contribution:** 3
**Rating:** 6
**Confidence:** 3

**Summary:**

The paper challenges the dominant narrative that using models with longer DNA sequences improves gene expression prediction (e.g., Enformer,AlphaGenome). The authors show that relatively short sequence lengths are often sufficient to preserve most of the predictive performance. They demonstrate that multimodal epigenomic signals (such as histone modifications and chromatin accessibility) carry rich cell-type--specific information. However, naïve integration of these signals introduces confounding effects, as background chromatin patterns correlate with gene expression. To address this issue, the authors propose a causal framework that learns multiple background chromatin states and applies backdoor adjustment to isolate true causal effects of regulatory signals. The paper reports improved gene expression prediction performance in two of the most well-characterized cell lines, outperforming the models it builds upon, i.e. Caduceus and Seq2Exp.

**Strengths:**

The paper presents a clear and well-supported argument that genomic sequence models do not significantly benefit from longer input sequences, a strong claim that is convincingly demonstrated through extensive experimentation (specifically Table 12). The results showing the impact of epigenetic markers are compelling and supported by thorough ablation studies that highlight the individual contribution of each signal type. The analysis of the confounding effect is insightful, and the proposed solution using backdoor adjustment is both conceptually well-suited to the biological problem, offering a principled way to disentangle causal relationships from spurious correlations.

**Weaknesses:**

My concerns regarding the efficiency of the proposed approach. As shown in the hyperparameter sensitivity analysis (Section 4.3), the variation in performance when tuning the parameters \alpha and \beta appears minimal, suggesting limited sensitivity to these design choices. Similarly, the number of background states n has only a minor impact on results, as even the case n=0 in Table 2a performs comparably well. This raises questions about how essential the proposed causal intervention mechanism truly is for achieving the reported improvements.

**Questions:**

- In the experimentation in Table 1, it is not clear to me the length of the sequences the model is using when computing the results in each row. Is Prism using a smaller genome sequence length when surpassing the SOTA models?

---

> ### Author Response · Authors · 2025-11-21
> **Response to Reviewer oNqu**
>
> Thank you for your thorough evaluation and constructive feedback. We address your concerns in detail below.
>
> > **W1:** Concerns regarding the efficiency of the proposed approach.
>
> When $n=0$, our method is equivalent to Caduceus w/signal [1], and one of our major findings is demonstrating that integrating multimodal epigenomic signals with short sequences already achieves strong performance. As shown in Table 1 below, Caduceus w/signal [1] with 2k input (equivalent to our $n=0$) performs comparably to Seq2Exp [2] with 200k input, validating our claim that Seq2Exp primarily mitigates the performance degradation caused by extending sequence length. In contrast, Prism takes an alternative approach: better multimodal signal integration achieves substantial improvements at a relatively optimal sequence length (2k input) for Caduceus.
>
> **Table 1**: Performance comparison on K562
>
> | Model | Input Length | MSE ↓ | MAE ↓ | Pearson ↑ |
> | --- | --- | --- | --- | --- |
> | Caduceus w/signal (equivalent to Prism $n=0$) | 2k | 0.1863 ± 0.0035 | 0.3092 ± 0.0050 | 0.8713 ± 0.0023 |
> | Seq2Exp-soft | 200k | 0.1856 ± 0.0032 | 0.3054 ± 0.0024 | 0.8723 ± 0.0012 |
> | Prism | 2k | **0.1789** ± 0.0041 | **0.3000** ± 0.0058 | **0.8751** ± 0.0036 |
>
> Regarding the other hyperparameters, we observe reasonable trends in the sensitivity analysis given the improvement magnitude Prism achieves over the $n=0$ baseline. It is notable that $\alpha$ shows a relatively significant effect. As $\alpha$ increases from 0 (equivalent to $n=0$) to 0.1 to 1.0, the MSE decreases from 0.1863 to 0.1829 to 0.1789, demonstrating clear improvement. The model's relative insensitivity to $\beta$ may be because end-to-end training naturally avoids overly collapsible weights, since such collapse would degrade the model to $n=1$, resulting in suboptimal performance. However, in practice, adding $\beta$ consistently yields better results than $n=1$.
>
> To further address your concern, we conducted additional experiments on the H1 cell line (which only appeared in Seq2Exp's rebuttal stage but was not included in their final camera-ready version). The results are shown in Table 2 below.
>
> **Table 2**: Performance on H1 cell line
>
> | Method | MSE ↓ | MAE ↓ | Pearson ↑ |
> | --- | --- | --- | --- |
> | EPInformer (reported by Seq2Exp rebuttal) | 0.2911 | 0.4005 | 0.6340 |
> | Caduceus w/signal (2k input) | 0.2751 ± 0.0104 | 0.3929 ± 0.0103 | 0.6681 ± 0.0137 |
> | Seq2Exp-soft (reported by Seq2Exp rebuttal) | 0.2724 | 0.3913 | 0.6684 |
> | Seq2Exp-soft (our reproduction) | 0.2784 ± 0.0064 | 0.3957 ± 0.0045 | 0.6595 ± 0.0089 |
> | Prism | **0.2642 ± 0.0060** | **0.3817 ± 0.0044** | **0.6844 ± 0.0078** |
>
> Notably, on H1, Seq2Exp performs worse than the Caduceus baseline, while Prism consistently achieves SOTA results across all metrics. These results on a third cell line further demonstrate the consistent improvements of our approach.
>
> > Q1: In the experimentation in Table 1, it is not clear to me the length of the sequences the model is using when computing the results in each row.
>
> All baseline results in Table 1 of our manuscript are directly taken from the reported results in the Seq2Exp [1] paper, and we explicitly mentioned the model sequence lengths in lines 371-372 of our manuscript. Specifically, in Table 1, except for EPInformer, which operates in a unique manner by only modeling annotated sub-regions within a 200k range, and Prism, which uses 2k bp, all other models use 200k sequences. Therefore, Prism indeed uses shorter sequences, which is precisely what we claim in our abstract and introduction. Additionally, we also reported the results of Caduceus w/signal with 2k input in Table 12 of our manuscript, which serves as a strong baseline.
>
> **References**:
>
> [1] Schiff et al., Caduceus: Bi-directional equivariant long-range DNA sequence modeling, ICML, 2024
>
> [2] Su et al., Learning to discover regulatory elements for gene expression prediction, ICLR, 2025

---

### Official Review · Reviewer_jLJ9 · 2025-11-01

**Soundness:** 2
**Presentation:** 3
**Contribution:** 3
**Rating:** 6
**Confidence:** 4

**Summary:**

**Problem (P)**

The paper is motivated by the issues that arises during multi-modal long-context genomic modelling. Specifically,
1. said models merely mitigate the performance degradation inherent in current long-sequence modelling paradigms. They do not fix it. Figure 1 d shows this.
2. the epigenomic signals are interdependent and play different roles. Naive modelling, with all of the signals, may render the model to learn spurious relations, the effects of which are demonstrated in figure 1 e/f.

**Solution (S)**

To mitigate the confounding effects Prism is introduced, an approach that:
1. learns high-dimensional feature combinations to represent background chromatin states, and
2. makes a prediction for each state and averages them (backdoor adjustment) to reduce confounding. Training uses three losses: prediction, intervention (on the averaged prediction), and a uniformity loss that promotes diversity in the learned weights.

**Contributions (C)**
1. The authors challenge the focus on long-sequence modeling for gene expression, showing it does not necessarily help with current tools.
2. They analyze roles of signals and point out that background chromatin patterns can confound models.
3. Introduce and evaluate Prism. The new method beats all baselines while operating at a 2kbp genomic context, while most baselines utilise 200kbp.

**Experimental Setting (E)**

1. Inputs: DNA sequence, sequence-wide H3K27ac, Hi-C, and DNase-seq data.
2. Baselines: Enformer, HyenaDNA, Mamba, Caduceus, EPInformer, Seq2Exp (hard and soft), Caduceus w/ signals, and MACS3 variant.
3. Metrics: MSE, MAE, Pearson.

**Strengths:**

1. Clear motivation **P**. The paper picks upon a prevalent issue in long-context DNA sequence modelling. The authors narrow down on the key-issue and validate it experimentally.
2. Within gene expression prediction, using latent background-state weights + uniform backdoor averaging is relatively novel.
3. While most baselines are trained and reported at 200k bp, Prism runs at 2k bp and still beats prior SOTA (Table 1). This supports their claim that better multi-modal integration can offset long-context modelling.
4. Ample baselines are explored and the evaluation metrics seem fine.

**Weaknesses:**

1. Prism completely discards long-range sequence information by design, operating on only 2kbp. This is presented as a strength, but I believe that this is also a fundamental limitation. The model cannot discover regulatory elements or sequence variations beyond its 2kbp window unless their effects are already captured by the provided proximal epigenomic signals. Have the authors explored how the metrics change when we increase the context? Why was 2k chosen?

2. Results in Table 1 are based on only two human cell lines, K562 and GM12878. While these are standard benchmarks, gene regulation is notoriously complex and cell-type specific. The model's SOTA performance may not hold across a wider, more diverse set of cell types or tissues. With the current breadth of exploration I feel that the proposed method has a limited experimental scope.

**Questions:**

Kindly address the weaknesses.

---

> ### Author Response · Authors · 2025-11-21
> **Response to Reviewer jLJ9**
>
> Thank you for your comments and helpful suggestions. We address your concerns in detail below.
>
> > W1: The model cannot discover regulatory elements or sequence variations beyond its 2kbp window unless their effects are already captured by the provided proximal epigenomic signals. Have the authors explored how the metrics change when we increase the context? Why was 2k chosen?
>
> Thank you for this important question. We have explored how performance changes with varying sequence lengths. In Table 1 below, we present the Pearson correlation results for Caduceus w/signal [1], Seq2Exp [2], and Prism on K562 across different input lengths ranging from 100 to 10,000 bp.
>
> **Table 1**: Pearson across different input lengths on K562
>
> | Input Length | Caduceus w/signal | Seq2Exp-soft | Prism ($\alpha$ = 1) | Prism ($\alpha$ = 2) |
> | --- | --- | --- | --- | --- |
> | 100 | 0.8488 ± 0.0042 | 0.8492 ± 0.0064 | **0.8493 ± 0.0056** | - |
> | 500 | 0.8719 ± 0.0043 | 0.8694 ± 0.0051 | **0.8726 ± 0.0045** | - |
> | 2000 | 0.8713 ± 0.0023 | 0.8643 ± 0.0088 | **0.8751 ± 0.0036** | - |
> | 5000 | 0.8662 ± 0.0035 | 0.8675 ± 0.0035 | **0.8690 ± 0.0037** | - |
> | 10000 | 0.8614 ± 0.0059 | **0.8699 ± 0.0032** | 0.8661 ± 0.0027 | **0.8699 ± 0.0023** |
>
> As shown in Table 1, Prism exhibits a similar trend to Caduceus, with performance beginning to decline when input length increases beyond 2000-5000 bp. While Seq2Exp maintains relatively robust performance across lengths, their reported results at 200k only match the performance of Caduceus at 500-2000 bp, validating our claim that Seq2Exp merely mitigates the performance degradation caused by extending sequence length in Caduceus. Prism achieves the best performance among all three models across lengths from 100 to 5000 bp. We hypothesize that as input length increases, the confounding effects of background signals become stronger. To test this, we experimented with increasing the intervention loss weight to $\alpha$ = 2 when the input length reaches 10,000 bp, and observed performance improvement.
>
> **Why 2k was chosen:** We selected 2k based on several considerations: (1) EPInformer [3] models the 2k proximal region as the promoter, which is biologically critical for gene expression; (2) Seq2Exp explicitly prevents masking of the central 2k region in its implementation; (3) Figure 1(d) in our manuscript demonstrates that 2k provides good performance. The results in Table 1 of this response further support our choice of 2k as an appropriate sequence length.
>
> **Interesting recent evidence**: Evo 1.5 [4], which was recently published in Nature, uses only 8k input length for better downstream task performance, despite Evo 1.0 [5] being pre-trained on 131k sequences. This further supports our observation that even genomic models like Evo, which primarily emphasize their long-context capabilities, have questionable ability to effectively process long sequences. Therefore, while we agree that in principle, very long context is needed to model elements such as distal enhancers, current models indeed cannot handle long sequences effectively.
>
> > **W2:** The model's SOTA performance may not hold across a wider, more diverse set of cell types or tissues. With the current breadth of exploration I feel that the proposed method has a limited experimental scope.
>
> Thank you for raising this concern. We would like to clarify our evaluation (including datasets) strictly follows the setup of previous work [2, 3]. To further address your concerns, we conducted additional experiments on the H1 cell line (which only appeared in Seq2Exp's rebuttal stage but was not included in their final camera-ready version). The results are shown in Table 2 below.
>
> **Table 2**: Performance on H1 cell line
>
> | Method | MSE ↓ | MAE ↓ | Pearson ↑ |
> | --- | --- | --- | --- |
> | EPInformer (reported by Seq2Exp rebuttal) | 0.2911 | 0.4005 | 0.6340 |
> | Caduceus w/signal (2k input) | 0.2751 ± 0.0104 | 0.3929 ± 0.0103 | 0.6681 ± 0.0137 |
> | Seq2Exp-soft (reported by Seq2Exp rebuttal) | 0.2724 | 0.3913 | 0.6684 |
> | Seq2Exp-soft (our reproduction) | 0.2784 ± 0.0064 | 0.3957 ± 0.0045 | 0.6595 ± 0.0089 |
> | Prism | **0.2642 ± 0.0060** | **0.3817 ± 0.0044** | **0.6844 ± 0.0078** |
>
> On H1, Seq2Exp performs worse than the Caduceus baseline, while Prism consistently achieves SOTA results across all metrics. These results on a third cell line further demonstrate the consistent improvements of our approach.
>
> **References**:
>
> [1] Schiff et al., Caduceus: Bi-directional equivariant long-range DNA sequence modeling, ICML, 2024
>
> [2] Su et al., Learning to discover regulatory elements for gene expression prediction, ICLR, 2025
>
> [3] Lin et al., EPInformer: a scalable deep learning framework for gene expression prediction, bioRxiv, 2024
>
> [4] Merchant et al., Semantic design of functional de novo genes from a genomic language model, Nature, 2025
>
> [5] Nguyen et al., Sequence modeling and design from molecular to genome scale with Evo, Science, 2024

---

### Official Review · Reviewer_a7M2 · 2025-11-01

**Soundness:** 3
**Presentation:** 3
**Contribution:** 3
**Rating:** 8
**Confidence:** 4

**Summary:**

This paper introduces Prism, a framework that accounts for the effects of confounders in epigenomic signals when predicting gene expression.  Specifically, it employs a three-layer 1D CNN as the confounder encoder, which processes the epigenomic signals to generate a set of confounder weight vectors.  In parallel, a single linear layer serves as the signal encoder, producing the corresponding signal weights. These two components are then combined, together with the DNA sequence features, to predict gene expression levels.

**Strengths:**

1. The introduction of confounder components for the gene expression prediction and their connection to biological intuition is important.
As it completes the current casual relationship formulation of the epigenomic signal.

2. The observation regarding the sequence length required for CAGE prediction is interesting and biologically reasonable.
The provided experiments support such observation on the K562 cell for Gene Expression CAGE Prediction.
I still have some doubts about whether a shorter sequence length is universally applicable to gene expression prediction tasks, or if it is specific to the datasets used in this study.
In other words, does 2k sequence is enough for all the gene expression prediction tasks beyond Gene Expression CAGE Prediction on the K562 and GM12878 cell?
If not, how to select the suitable length for diverse task?

3. The experimental results looks good with the introduction of the confounder components. Overall, the paper writing is clear.

**Weaknesses:**

The overall framework appears well designed and complete, and I have no further comments regarding potential improvements.
My remaining concern lies in how to determine the appropriate sequence length for different prediction tasks.
Furthermore, if the goal is to train a unified model for general gene expression prediction, it would be helpful to clarify how the model can adapt to varying sequence length requirements across different genes or datasets.

**Questions:**

N/A

---

> ### Author Response · Authors · 2025-11-21
> **Response to Reviewer a7M2 （Part 1/2)**
>
> Thank you for your comments and helpful suggestions. We address your concerns in detail below.
>
> > **W1:** Is 2k sequence enough for all gene expression prediction tasks beyond CAGE prediction on K562 and GM12878? If not, how to select the suitable length for diverse tasks?
>
> Thank you for this important question. Determining the optimal input sequence length for a specific task is challenging in genomic deep learning, as it depends on both the model's capacity to process sequences of varying lengths and the inherent characteristics of the dataset and task. Recent genomic foundation models, including NT [1], Evo [2], Enformer [3], and Borzoi [4], all adopt fixed training lengths, with most evaluating at the same length used during training. Similarly, gene expression prediction methods such as EPInformer [5] and Seq2Exp [6] also use fixed sequence lengths throughout training and evaluation.
>
> We selected 2k as our input length based on several considerations: (1) EPInformer [5] models the 2k proximal region as the promoter, which is critical for gene expression; (2) Seq2Exp [6] explicitly prevents masking of the central 2k region in its implementation; (3) Figure 1(d) in our manuscript demonstrates that 2k provides strong performance. However, we acknowledge that 2k may not be universally optimal across all gene expression prediction tasks, and the ideal length should be validated for specific datasets through systematic experiments.
>
> **Interesting recent evidence**: Evo 1.5 [7], which was recently published in Nature, uses only 8k input length for better downstream task performance, despite Evo 1.0 [2] being pre-trained on 131k sequences. This further supports our observation that even genomic models like Evo, which primarily emphasize their long-context capabilities, have questionable ability to effectively process long sequences. This also highlights the difficulty of selecting the optimal sequence length during the training phase.
>
> > **W2:** It would be helpful to clarify how the model can adapt to varying sequence length requirements across different genes or datasets.
>
> Thank you for this suggestion. We address this in two parts: cross-cell generalization and sequence length adaptation.
>
> **Cross-cell generalization:** Our main experiments follow EPInformer [5] and Seq2Exp [6] in training separate models for each cell type. To evaluate whether a single model can generalize across cell types, we conducted a mixed-training experiment combining K562 and GM12878. Specifically, during training, for each gene we randomly sample either its K562 or GM12878 epigenomic signals and corresponding expression value. The Table 1 below shows that this mixed model achieves comparable performance to cell-type-specific models:
>
> **Table 1**: Performance of mixed-training
> | Dataset | Model | MSE ↓ | MAE ↓ | Pearson ↑ |
> | --- | --- | --- | --- | --- |
> | K562 | Seq2Exp (cell-specific) | 0.1856 ± 0.0032 | 0.3054 ± 0.0024 | 0.8723 ± 0.0012 |
> | K562 | Prism (cell-specific) | 0.1789 ± 0.0041 | 0.3000 ± 0.0058 | 0.8751 ± 0.0036 |
> | K562 | Prism (mixed-training) | 0.1875 ± 0.0085 | 0.3084 ± 0.0077 | 0.8662 ± 0.0049 |
> | GM12878 | Seq2Exp (cell-specific) | 0.1873 ± 0.0044 | 0.3137 ± 0.0028 | 0.8951 ± 0.0038 |
> | GM12878 | Prism (cell-specific) | 0.1759 ± 0.0054 | 0.3054 ± 0.0048 | 0.9016 ± 0.0024 |
> | GM12878 | Prism (mixed-training) | 0.1759 ± 0.0038 | 0.3027 ± 0.0041 | 0.9012 ± 0.0032 |
>
>
> The mixed-training model performs comparably to Seq2Exp on K562 and maintains near-identical performance on GM12878, demonstrating cross-cell generalization capability.
>
>
> **Sequence length adaptation:** As discussed in W1, determining optimal length during training is challenging. However, models trained on longer sequences often generalize well to shorter inputs. As shown in Figure 2 of our manuscript, Seq2Exp trained on 200k sequences maintains performance even when tested with inputs shortened to 2.5k. This suggests a practical strategy: (1) pre-train on a moderately long sequence length (e.g., 10k) across multiple cell types; (2) when targeting specific cell types or tasks, fine-tune and validate with different sequence lengths to identify the task-specific optimum. This strategy is similar to Evo 1.5 [7] and may identify the optimal sequence length for a specific dataset with relatively minimal additional computational overhead.

---

> ### Author Response · Authors · 2025-11-21
> **Response to Reviewer a7M2 （Part 2/2)**
>
> **References**:
>
> [1] Dalla-Torre et al., The Nucleotide Transformer: Building and Evaluating Robust Foundation Models for Human Genomics, Nature Methods, 2024
>
> [2] Nguyen et al., Sequence modeling and design from molecular to genome scale with Evo, Science, 2024
>
> [3] Avsec et al., Effective gene expression prediction from sequence by integrating long-range interactions, Nature Methods, 2021
>
> [4] Linder et al., Predicting RNA-seq coverage from DNA sequence as a unifying model of gene regulation, Nature Genetics, 2025
>
> [5] Lin et al., EPInformer: a scalable deep learning framework for gene expression prediction, bioRxiv, 2024
>
> [6] Su et al., Learning to discover regulatory elements for gene expression prediction, ICLR, 2025
>
> [7] Merchant et al., Semantic design of functional de novo genes from a genomic language model, Nature, 2025

---

### Author Response · Authors · 2025-12-01
**Discussion Summary**

We would like to thank all reviewers for **their consistently positive scores** on our paper **during the initial assessment** (Scores: 8, 6, 6, 6; Avg: 6.5). We also extend our gratitude to the ACs for their efforts in reviewing our work.

We sincerely appreciate all reviewers for their constructive feedback, which has significantly improved the quality of our paper. During the rebuttal period, we provided detailed responses and additional experiments to address the reviewers' concerns. **No reviewers raised further comments or adjusted their scores** before the discussion was unexpectedly closed. We have incorporated all relevant experimental results into **Appendix I-L of the revised manuscript**. Below, we summarize the key concerns raised by the reviewers along with our responses.

> 1: Input sequence length selection (Reviewers `a7M2` and `jLJ9`)

We first provided detailed justification for our choice of 2k input length, which was primarily based on previous literature rather than exhaustive optimization. We then supplemented experiments comparing Prism with baselines Caduceus and Seq2Exp across varying sequence lengths (Table 1 in response to `jLJ9`). Results show that, similar to Caduceus, Prism's performance begins to decline when input length exceeds approximately 5k bp, though this decline can be mitigated by adjusting other hyperparameters. Finally, we demonstrated through mixed-training experiments (Table 1 in response to Reviewer `a7M2`) that training across multiple cell types incurs minimal performance loss. This suggests an efficient training strategy: pre-train on a moderately long sequence length across multiple cell types, then fine-tune to identify the cell-type-specific optimal length with minimal additional computational cost.


> 2: Limited experimental scope (Reviewer `jLJ9`)

We clarified that our evaluation strictly follows the protocol established by prior work, EPInformer and Seq2Exp. We expanded our evaluation to a third cell line, H1, as shown in Table 2 (response to `jLJ9`). On the H1 cell line, Prism consistently outperforms both Seq2Exp and Caduceus across all metrics, confirming that our SOTA performance generalizes across diverse cell types.

> 3: Prism's efficiency (Reviewer `oNqu`)

We clarified that when $n=0$, Prism is equivalent to Caduceus w/signal, which aligns with one of our major findings: integrating multimodal epigenomic signals with short sequences already achieves strong performance. Beyond this strong baseline, Prism demonstrates notable improvements. Additionally, we supplemented results on the H1 cell line, demonstrating Prism's consistent and robust improvements across different datasets.

> 4: How the model can learn meaningful weights (Reviewer `uN9e`)

We first demonstrate through existing work that unsupervised methods can learn meaningful chromatin states. To validate that our approach differs fundamentally from naive data augmentation, we conducted Caduceus dropout baseline experiments showing that random dropout causes significant performance degradation, whereas Prism's learned weights maintain superior performance despite having a lower average retention rate. This demonstrates that the learnable weights are meaningful, selectively preserving essential biological information rather than applying random perturbations. Additionally, the visualization in Figure 5 of our manuscript provides further evidence that the learned weights capture structured, gene-specific patterns. Finally, we addressed Reviewer `uN9e`'s open question by proposing a training scheme for supervised settings.

---

### Meta-Review · Area_Chair_FMtw · 2025-12-16

**Summary:**

Reviewers consistently agree that this paper addresses a notable problem in gene expression prediction, i.e., challenging the prevailing assumption that long-sequence modeling uniformly yields performance benefits. The proposed solution, which leverages backdoor adjustment to model confounder effects, is conceptually well-aligned with the biological problem, and demonstrates a degree of methodological novelty. Moreover, the authors’ core claim is rigorously substantiated through comprehensive experiments, by systematically comparing baseline methods on longer contexts against their proposed approach on shorter contexts.

In conclusion, the paper’s well-validated findings are of considerable interest to the field and have the potential to reshape the traditional paradigm of sequence modeling for gene expression prediction tasks.

**Reviewer Concerns:**

As summarized in the authors' response, some detailed technical problems have been addressed during the rebuttal, including sequence selection, efficiency, and parameter settings.

**Reviewer Scores:**

Since all reviewers have already given very high scores (8666), I don't think they will further raise their scores.

---

### Decision · Program_Chairs · 2026-01-26

Accept (Oral)